# Flexible Realignment of Language Models

Wenhong Zhu[1,2]      Ruobing Xie[3]      Weinan Zhang[1,2]      Rui Wang[1]*

[1]Shanghai Jiao Tong University    [2]Shanghai Innovation Institute
[3]Large Language Department, Tencent

## Abstract

Realignment becomes necessary when a language model (LM) fails to meet expected performance. We propose a flexible realignment framework that supports quantitative control of alignment degree during training and inference. This framework incorporates **Training-time Realignment (TrRa)**, which efficiently realigns the reference model by leveraging the controllable fusion of logits from both the reference and already aligned models. For example, TrRa reduces token usage by **54.63%** on DeepSeek-R1-Distill-Qwen-1.5B without any performance degradation, outperforming DeepScaleR-1.5B's **33.86%**. To complement TrRa during inference, we introduce a layer adapter that enables **smooth Inference-time Realignment (InRa)**. This adapter is initialized to perform an identity transformation at the bottom layer and is inserted preceding the original layers. During inference, input embeddings are simultaneously processed by the adapter and the original layer, followed by the remaining layers, and then controllably interpolated at the logit level. We upgraded DeepSeek-R1-Distill-Qwen-7B from a slow-thinking model to one that supports both fast and slow thinking, allowing flexible alignment control even **during inference**. By encouraging deeper reasoning, it even surpassed its original performance.

## 1 Introduction

Current large language models (LLMs), such as GPT-4o [1], and reasoning-focused models like OpenAI-o1 [2] and DeepSeek-R1 [3], have achieved remarkable success. These models typically hinge on a series of critical training phases [4]. First, they undergo pre-training on vast corpora to master the ability to predict the next token [5]. Next, the pre-trained models are fine-tuned through supervised fine-tuning (SFT) as a *cold start* to better adapt to specific domains [6, 7]. Reinforcement Learning (RL) has emerged as a crucial component of the entire training pipeline.

In the RL phase, the core objective is to maximize the expected reward while incorporating the KL-divergence from the reference policy [8–10]. The reward signal is key to **alignment**: correctness-based rewards improve reasoning ability [2, 3], while 3H-based (honesty, harmlessness, and helpfulness) rewards reflect human values [11]. However, misalignment can still emerge due to imperfect rewards or evolving user needs. Review the pain points of the existing product models: The most advanced reasoning models tend to suffer from the overthinking problem [12, 3], leading to increased computational costs. How can we realign these models for efficient reasoning to ensure user affordability? Meanwhile, to adapt to individual user preferences, conversational models often become overly sycophantic [13]. How can we realign them to balance personalization and objective responses better? **Realignment** is thus essential to correct model behavior and ensure robustness over time.

---

*Corresponding author. Code: https://github.com/zwhong714/ReAligner    Email: zwhong714@sjtu.edu.cn

39th Conference on Neural Information Processing Systems (NeurIPS 2025).

One typical practical approach to realignment is to retrain the model under the same reward signal while exploring different hyperparameters. However, for models trained via RL, this process is often resource-intensive. For instance, simply replicating the DeepSeek-R1 experiments (with context lengths exceeding 32K over 8000 training steps) using a 1.5B-parameter model requires at least 70,000 A100 GPU hours[14]. We need to address this challenge through a more efficient method without compromising performance. Additionally, we seek a solution that offers flexibility during training and inference.

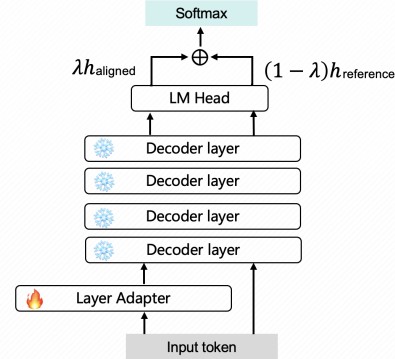

Figure 1: Our **InRa**: The inputs are fed simultaneously into the layer adapter and the original bottom layer of the LM. The hidden states from both paths are propagated through all layers and merged at the logit level. The layer adapter enables flexible realignment even during inference.

We propose a **flexible realignment framework** that facilitates training-time and inference-time realignment, controlling the alignment degree to satisfy different demands. (1) We draw inspiration from knowledge distillation for **training-time realignment (TrRa)**. Specifically, we realign the reference model using a teacher signal constructed from a controllable fusion of the output logits of the reference and the already aligned models. (2) We introduce a layer adapter to endow the LM **inference-time realignment (InRa)**. This is inspired by the fact that the lower layers of the LM are more influential than the upper layers during fine-tuning (see Sec.5.1). Based on this observation, we duplicate the bottom layer and insert it as an identity mapping layer before the original layers (see Sec.3.2). Fine-tuning is restricted solely to this layer adapter. As illustrated in Figure 1, the input embeddings are processed through this adapter and original layers during inference, and the resulting output logits are combined via an interpolation coefficient $\lambda$. This design retains both paths of logits within a single model, enabling smooth and flexible control over alignment.

In summary, our main contributions are as follows:

- We propose new post-training methods, TrRa and TrRa-iter, that use a controllable teacher signal created by combining logits from reference and aligned models, overcoming the fixed teacher in traditional knowledge distillation to enable flexible training-time realignment.

- We propose a parameter-efficient fine-tuning approach called the layer adapter. It allows smooth and efficient realignment during inference within a single model.

- Experiments demonstrate that our flexible realignment framework enables efficient and flexible realignment during both training and inference. For example, TrRa-iter reduces token usage by 54.63% on DeepSeek-R1-Distill-Qwen-1.5B without any loss in performance. Additionally, InRa has been successfully tested in practical scenarios, such as combining fast and slow thinking models and flexibly realigning with 3H values.

## 2    Preliminary

**Autoregressive LM.** Given a query sequence $x := (x_1, \ldots, x_m) \in \mathcal{X}$, an auto-regressive LM defines a probability distribution over possible response sequences $y := (y_1, y_2, \ldots, y_n) \in \mathcal{Y}$. The probability $\pi_\theta(y \mid x)$ can be decomposed using the chain rule of probability as $\pi_\theta(y \mid x) = \prod_{t=1}^{n} \pi_\theta(y_t \mid y_{<t}, x)$, where $y_{<t}$ denotes $\{y_1, y_2, ..., y_{t-1}\}$.

**Transformer Decoder Layer.** The current mainstream LMs based on transformer architecture [15] typically have multiple decoder layers $(\phi_0, \phi_1, ..., \phi_L)$ [4]. Each layer consists of an attention component and an MLP component. Given an input $h_{t-1}$, the layer computes the output $h_t$ through the following steps: $h'_{t-1} = h_{t-1} + \text{Attention}(\text{RMSNorm}(h_{t-1}))$ and $h_t = h'_{t-1} + \text{MLP}\left(\text{RMSNorm}\left(h'_{t-1}\right)\right)$. Both components have a projector to ensure the module's input and output dimensions are consistent, facilitating the combination with a residual connection [16].

**Reward-based Fine-tuning.** Given a pre-trained (Base) and typically SFT **reference model** $\pi^{\text{ref}}(y \mid x)$, RL is a commonly used post-training technique to enhance model capabilities further. The

optimization objective maximizes the expected reward $r(x, y)$ while including a KL-divergence term from the reference policy as a penalty. The objective is as follows:

$$\max_{\pi_\theta} \mathbb{E}_{x \sim \mathcal{X}, y \sim \pi_\theta(y|x)} \left[ r(x, y) - \beta \log \frac{\pi_\theta(y \mid x)}{\pi^{\text{ref}}(y \mid x)} \right], \tag{1}$$

where $\beta$ is a regularization parameter. It has a closed-form solution for the **aligned model**, given as follows:

$$\pi_\theta^*(\beta)(y \mid x) = \frac{\pi^{\text{ref}}(y \mid x) \exp\left[\frac{1}{\beta} r(x, y)\right]}{\sum_{y'} \pi^{\text{ref}}(y' \mid x) \exp\left[\frac{1}{\beta} r(x, y')\right]}. \tag{2}$$

Typically, we can transform the above equation by representing $r(x, y)$ as

$$\frac{1}{\beta} r(x, y) = \log \frac{\pi_\theta^*(\beta)(y \mid x)}{\pi^{\text{ref}}(y \mid x)} + \log Z(x), \tag{3}$$

where $Z(x) := \sum_{y'} \pi^{\text{ref}}(y' \mid x) \exp\left(\frac{1}{\beta} r(x, y')\right)$ is the partition function.

**Realignment.** Realignment becomes necessary when the LM fails to meet expected performance. As shown in Equation 1, the KL regularization parameter $\beta$ determines how far the policy model $\pi_\theta(y \mid x)$ deviating from its initial state $\pi^{\text{ref}}(y \mid x)$ [17]. To adjust the alignment strength of the LM, one can modify the value of $\beta$, which can be achieved by scaling it with a factor $\lambda$ during training. This adjustment leads to an updated optimal solution for the **realigned model**, expressed as $\pi_\theta^*(\beta/\lambda)(y \mid x)$ as follows:

$$\pi_\theta^*(\beta/\lambda)(y \mid x) = \frac{\pi^{\text{ref}}(y \mid x) \left[\frac{\pi_\theta^*(\beta)(y|x)}{\pi^{\text{ref}}(y|x)}\right]^\lambda}{\sum_{y'} \pi^{\text{ref}}(y' \mid x) \left[\frac{\pi_\theta^*(\beta)(y'|x)}{\pi^{\text{ref}}(y'|x)}\right]^\lambda}. \tag{4}$$

However, computing Equation 4 is infeasible due to the normalization constant involving all possible sequences. DeRa [18] demonstrates that Equation 4 can be approximated at the per-token level through the auto-regressive property of LMs. When decoding token by token, it combines the logits from the reference model, $\boldsymbol{h}_t^{\text{ref}}$, with its from the $\beta$-regularization-aligned model, $\boldsymbol{h}_t^\theta(\beta)$, at each time step $t$, as detailed below.

$$\widehat{\pi}_\theta(\beta/\lambda)(\cdot \mid x, y_{<t}) := \text{softmax}\left[\lambda \boldsymbol{h}_t^\theta(\beta) + (1 - \lambda)\boldsymbol{h}_t^{\text{ref}}\right]. \tag{5}$$

The interpolation parameter $\lambda$ functions as readjusting the alignment strength during the inference. Refer to Appendix C.1 for the proof.

## 3 Flexible Realignment Framework

This section introduces our flexible realignment framework, which enables LM realignment during training and endows the LM with the capability of dynamic realignment during inference.

### 3.1 Training-time Realignment

Based on the previous description, DeRa [18] approximately doubles the decoding time and memory consumption. However, it reveals an appealing property: the reference and aligned models can be interpolated at the logit level. Drawing inspiration from knowledge distillation, where the student model learns from the teacher's predicted probability distribution, we propose an innovative approach where Equation 5 serves as the teacher's distribution to realign the reference model. Our method minimizes the following objective function:

$$\mathcal{L} = D_{\text{KL}}(\pi_\theta(\cdot|x, y_{<t}) || \hat{\pi}_\theta(\beta/\lambda)(\cdot|x, y_{<t})). \tag{6}$$

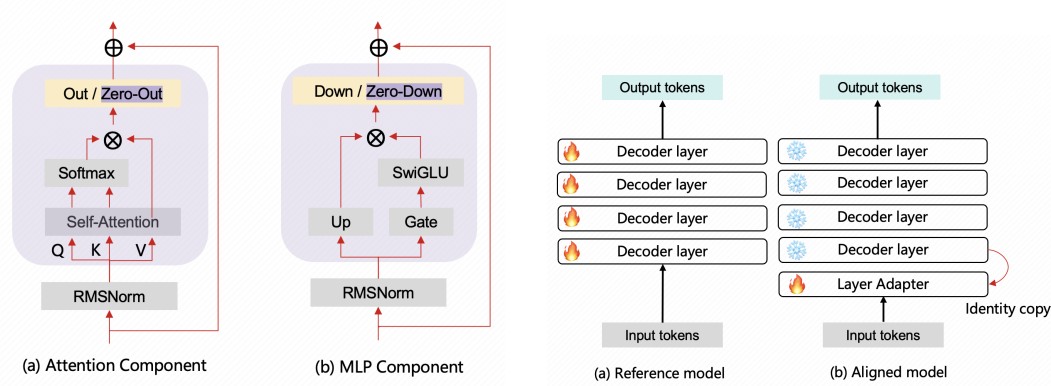

Figure 2: Overview of attention and MLP component. Identity copy makes the last projector of each component with weight and bias to zero.

Figure 3: (a) All layers fine-tuning. (b) Fine-tuning on the added identity layer while keeping the original layers of the LM frozen.

**Flexible Control During Training**    In previous distillation approaches, the distribution of teachers typically remained fixed. However, TrRa introduces the capability to dynamically generate multiple teacher distributions by flexibly adjusting $\lambda$. These teacher distributions are derived from reference and aligned models, enabling the interpolation and extrapolation of the reward signal.

**Iterative Realignment**    We can apply TrRa iteratively (TrRa-iter) to derive a better model. Suppose $\mathbb{A}$ denotes the base model and $\mathbb{B}$ the aligned model. A realigned model $\mathbb{C}$ can be derived using Objective 6. This procedure can be iteratively extended to $\mathbb{A}$ and $\mathbb{C}$, resulting in a further realigned model.

## 3.2    Inference-time Realignment

To complement TrRa, we aim to equip the LM with realignment capability during inference, enabling more flexible use for end users. We duplicate the original LM's bottom layer as an identity copy and insert it before the original layers. The layer adapter can be fine-tuned to inject alignment information, such as short-thinking patterns or 3-H values.

**Identity copy**    The *identity copy* is defined as $\phi_{\mathrm{id}}(h_{t-1}) = h_{t-1}$, which means the input and output are identical. This can be achieved as long as Attention(RMSNorm($h_{t-1}$)) = 0 and MLP(RMSNorm($h'_{t-1}$)) = 0. Then, the input is directly the result of the output due to the residual. We initialize the projection weight matrices—$W_{\mathrm{out}}$ in the Attention module and $W_{\mathrm{down}}$ in the MLP—as indicated by the dark purple regions in Figure 2, setting them to zero to ensure this identity property.

**Layer adapter**    *Layer expansion* involves inserting additional layers into the original layer structure. Incorporating identity layers ensures that the added layers do not compromise the original capabilities of LMs. As illustrated in Figure 3, we duplicate the bottom layer from the original model and insert it as an identical mapping. LoRA is orthogonal to our method. The principle of LoRA [19] is given by $W_0 + \Delta W = W_0 + BA$, where $A$ is initialized with $\mathcal{N}\left(0, \sigma^2\right)$ and $B$ is a zero matrix. Therefore, LoRA is also initialized as an identity component. We can implement our method using trainable rank decomposition matrices.

**Training**    As shown in Figure 3, we freeze the original layers of the LMs and perform fine-tuning only on the layer adapter. This training guarantees fine-tuning starting from the original distribution. **The critical aspect of this step is the injection of the reward signal.**

**Inference**    The decoding architecture is depicted in Figure 1. During the inference phase, the LM processes the input embeddings by passing them through the layer adapter alongside the original bottom layer of the LM. The **hidden states** generated by both layers are retained and subsequently fed into the remaining layers. The aligned and reference **logits** are combined in the LM head layer

Table 1: Performance comparison of different models on three benchmarks. 'iter + $x$' denotes iterative realignment applied $x$ times under the same settings. Red indicates improved performance compared to DeepScalerR-1.5B-Preview, while green indicates a decrease.

| Models | AIME24 | | AIME25 | | MATH-500 | | Token Reduction% |
|---|---|---|---|---|---|---|---|
| | Pass@1 | #Token | Pass@1 | #Token | Pass@1 | #Token | |
| DeepSeek-R1-Distill-Qwen-1.5B | 30.00 | 12602 | 19.58 | 12278 | 80.23 | 4699 | – |
| DeepSeek-R1-TrRa-1.5B-$\lambda = 0.5$ | 38.33 | 10678 | 28.75 | 10254 | 83.70 | 3734 | 17.42 |
| DeepScaleR-1.5B-Preview | 37.50 | 8520 | 30.41 | 8143 | 85.20 | 3030 | 33.86 |
| DeepSeek-R1-TrRa-1.5B-$\lambda = 1.5$ | 41.25 | 8091 | 30.83 | 7353 | 85.20 | 2982 | 37.48 |
| DeepSeek-R1-TrRa-1.5B-$\lambda = 2.0$ | 37.50 | 7441 | 28.33 | 6498 | 84.98 | 2897 | 42.11 |
| DeepSeek-R1-TrRa-1.5B-$\lambda = 5.0$ | 31.25 | 6297 | 27.50 | 5652 | 83.95 | 2844 | 47.83 |
| DeepSeek-R1-TrRa-1.5B-$\lambda = 10.0$ | 29.58 | 6004 | 25.00 | 5174 | 81.58 | 2713 | 50.83 |
| DeepSeek-R1-TrRa-iter1-1.5B-$\lambda = 2.0$ | 29.17 | 5631 | 25.00 | 4434 | 81.35 | 2599 | 54.63 |
| DeepSeek-R1-TrRa-iter2-1.5B-$\lambda = 2.0$ | 14.58 | 4294 | 15.42 | 3887 | 75.93 | 2483 | 60.48 |

using the interpolation parameter $\lambda$. This parameter functions similarly to a temperature setting, enabling users to customize the desired alignment strength **smoothly**. Merging hidden states in early layers can hurt model performance, often causing repeated outputs like "!!!!!".

### 3.3 Discussions on Training/Inference-time Realignment

TrRa and InRa are orthogonal. (1) TrRa realigns the model during training, ensuring flexibility and performance. In contrast, InRa also requires training; however, its training phase focuses on injecting the reward signal into the layer adapter, and this training can be done via SFT, DPO, or TrRa. See Appendix D for justification. (2) InRa retains both aligned and reference logits, enabling realignment during inference. This feature incurs additional KV-cache storage overhead. Although we integrate InRa into the vLLM framework [20], it leads to a decrease in inference throughput. See Appendix A for potential solutions.

## 4 Experiments

In Section 4.1, we demonstrate the effectiveness of TrRa during training. Section 4.2 presents an extension of the current slow-thinking model into a slow-fast thinking framework. In Section 4.3, we explore the integration of 3H-values into the chatbot model. In the latter two sections, we evaluate the effects of realignment during inference.

### 4.1 Training-time Realignment

**Evaluation Settings.** (a) *Models and Baselines:* We use DeepSeek-R1-Distill-Qwen-1.5B[3] as our reference model and DeepScaleR-1.5B-Preview (trained on 40K high-quality math problems with 3,800 A100 hours)[14] as our aligned model. (b) *Calibrated Training Datasets:* We use the OpenR1-Math-220K dataset [21]. Due to computational constraints, we filter samples with generation lengths between 4k and 8k. (c) *Evaluation Dataset:* We evaluate on challenging reasoning tasks including AIME-24, AIME-25, and MATH-500 to assess performance. (d) *Setup:* We realign DeepSeek-R1-Distill-Qwen-1.5B for 200 steps with a batch size of 16. Performance is measured using the Pass@1 metric and token count, where we sample 8 generations per example and report the average score. Each generation has a maximum length of 16384 tokens, with temperature set to 0.7 and top-p set to 0.95.

**Results.** As shown in Table 1, DeepScaleR-1.5B-Preview demonstrates strong performance and efficient reasoning. By applying our TrRa method to realign DeepSeek-R1-Distill-Qwen-1.5B, we make the following observations:

(a) ***TrRa is an efficient and flexible alignment controller.*** It can be seen that we achieve effective realignment at a very low cost compared to the training cost of DeepScaleR. With correction applied using a context length of just 4k to 8k, the model generalizes well to 16k during inference. Besides, we can control the degree of alignment achieved by sweeping over different values of $\lambda$.

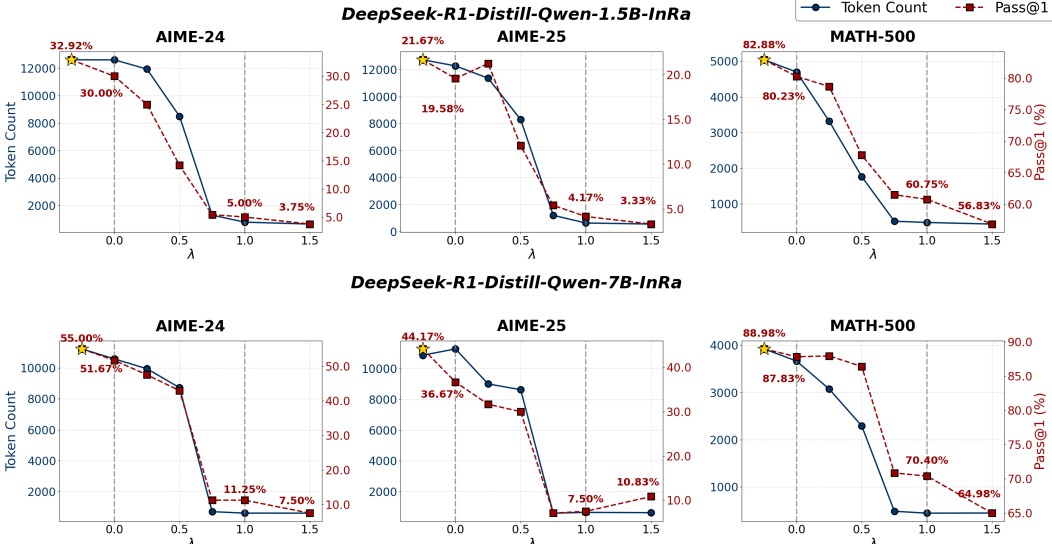

Figure 4: Reasoning Performance on different models and benchmarks with our InRa, verifying the successful interpolation and extrapolation of realignment. $\lambda = 0$ means merely using slowing thinking, while $\lambda = 1$ indicates solely using fast thinking.

(b) **TrRa leads to a more efficient reasoning pattern.** By appropriately increasing the value of $\lambda$, the reasoning becomes concise without sacrificing correctness. Notably, even at $\lambda = 10$, the model achieves a 50.8% reduction in tokens while outperforming DeepSeek-R1-Distill-Qwen-1.5B.

(c) **TrRa-iter can further amplify this efficient reasoning pattern.** Iterative realignment results in more efficient reasoning than setting a large initial $\lambda$. However, as the iteration of realignments increases, the model tends to produce more concise reasoning at the cost of lower correctness.

## 4.2 Inference-time Realignment for Reasoning

This section explores integrating fast and slow thinking modes into a unified model. Within this model, a floating-point hyperparameter like `temperature` can **smoothly** adjust the balance between the two modes of thinking.

**Evaluation Settings.** (a) *Models and Baselines:* We adopt DeepSeek-R1-Distill-Qwen-1.5B [3] and DeepSeek-R1-Distill-Qwen-7B [3] as our primary models. (b) *Training Datasets:* We perform SFT on short CoT segments from the OpenR1-Math-220K dataset (after the `</think>` tag), yielding controllable reasoning models named by adding the `-InRa` suffix to the original models. (c) *Setup:* We train our model for three epochs using a batch size of 128. (d) *Evaluation:* The evaluation setting is the same as in Section 4.1.

**Results.** The results are shown in Figure 4. We provide the following analyses:

(a) **The degree of alignment could be flexibly adjusted even AFTER training.** It confirms the practicality of our proposed InRa.

(b) **InRa enables continuous transformation of reasoning tokens by tuning the realignment parameter $\lambda$.** Generally, the extent of reasoning significantly changes around $\lambda = 0.5$. For detailed examples, see Appendix F.7.

(c) **Extrapolation further enhances model performance.** When $\lambda > 1$, it encourages the model to use fast thinking. We can see that fasting thinking sacrifices reasoning accuracy with the token decreasing. Surprisingly, we even found that $\lambda < 0$ may encourage the model to **think more and perform better.** Notably, the reasoning accuracy on all three benchmarks exceeds that of the original reasoning model (e.g., DeepSeek-R1-Distill-Qwen-7B), with nearly **4%** average improvements on AIME-24, AIME-25, and MATH-500.

Table 2: Evaluation results of models across different settings and benchmarks. LC and WR refer to length-controlled and raw win rates, respectively. At $\lambda = 0.0$, the model corresponds to the original SFT version, while at $\lambda = 1.0$, it represents our efficient fine-tuning method using DPO. Red indicates improved performance compared to DPO full fine-tuning, green indicates a decrease.

| Method | Llama3.2-3B-Base | | | | | Llama3.1-8B-Base | | | | |
| | AlpacaEval2 | | Arena-Hard | MT-Bench | | AlpacaEval2 | | Arena-Hard | MT-Bench | |
| | LC (%) | WR (%) | WR (%) | 1-turn | 2-turn | LC (%) | WR (%) | WR (%) | 1-turn | 2-turn |
|---|---|---|---|---|---|---|---|---|---|---|
| SFT | 2.83 | 2.42 | 2.00 | 6.39 | 5.53 | 3.06 | 2.79 | 3.60 | 6.99 | 6.43 |
| DPO$_{Full}$ | 11.44 | 10.47 | 11.60 | 6.98 | 6.51 | 20.16 | 15.02 | 26.20 | 7.66 | 7.28 |
| DeRa$_{\lambda=2.0}$ | 4.50 | 4.97 | 19.80 | 6.30 | 5.30 | 28.63 | 25.68 | 36.30 | 7.74 | 7.45 |
| InRa$_{\lambda=0.5}$ | 7.28 | 6.62 | 6.90 | 6.66 | 6.18 | 12.19 | 9.80 | 14.00 | 7.26 | 7.03 |
| InRa$_{\lambda=1.0}$ | 11.53 | 11.93 | 11.00 | 7.11 | 6.36 | 20.45 | 17.86 | 24.40 | 7.46 | 7.13 |
| InRa$_{\lambda=1.5}$ | 12.71 | 14.84 | 17.30 | 6.88 | 6.74 | 21.99 | 20.81 | 29.80 | 7.67 | 7.31 |
| InRa$_{\lambda=2.0}$ | 12.01 | 14.92 | 21.80 | 6.93 | 6.24 | 20.90 | 21.09 | 34.30 | 7.36 | 7.04 |

| Method | Qwen2.5-1.5B-Base | | | | | Qwen2.5-7B-Base | | | | |
| | AlpacaEval2 | | Arena-Hard | MT-Bench | | AlpacaEval2 | | Arena-Hard | MT-Bench | |
| | LC (%) | WR (%) | WR (%) | 1-turn | 2-turn | LC (%) | WR (%) | WR (%) | 1-turn | 2-turn |
|---|---|---|---|---|---|---|---|---|---|---|
| SFT | 2.55 | 2.54 | 2.80 | 6.85 | 5.30 | 5.42 | 3.59 | 8.30 | 7.43 | 7.01 |
| DPO$_{Full}$ | 8.38 | 8.36 | 15.60 | 7.49 | 6.68 | 25.20 | 20.92 | 45.50 | 8.21 | **7.80** |
| DeRa$_{\lambda=2.0}$ | 4.50 | 4.97 | 9.90 | 7.09 | 6.59 | 36.53 | 34.73 | 58.00 | 8.59 | 8.28 |
| InRa$_{\lambda=0.5}$ | 4.75 | 5.04 | 7.00 | 6.98 | 6.13 | 14.19 | 11.52 | 30.50 | 8.17 | 7.53 |
| InRa$_{\lambda=1.0}$ | 9.13 | 10.50 | 14.80 | 7.51 | 6.59 | 25.74 | 25.83 | 45.10 | 8.21 | 6.70 |
| InRa$_{\lambda=1.5}$ | 8.64 | 11.23 | 16.30 | 7.48 | 6.54 | 30.69 | 34.15 | 50.80 | 8.48 | 5.94 |
| InRa$_{\lambda=2.0}$ | 7.15 | 10.38 | 15.30 | 7.58 | 6.68 | 30.99 | 35.35 | 50.30 | 8.51 | 5.58 |

## 4.3 Inference-time Realignment for Dialogue Model

This section explores the 3H-values realignment in dialogue models. The GPT-4o sycophancy incident [13] on April 25th, 2025, highlights the importance of balancing the reward signals in dialogue systems, and thus a flexible alignment controller is desired.

**Evaluation Settings.** (a) *Models:* We implement our proposed method on the Llama3.2-3B [22], Llama3.1-8B [22], Qwen2.5-1.5B [23] and Qwen2.5-7B models [23]. (b) *Baselines:* Full fine-tuning using **DPO** [10] and the DeRa method [18]. (d) *Setup:* We first train the base models using the UltraChat-200k dataset [24], which contains 1.5 million high-quality multi-turn dialogues, to obtain the SFT models. Subsequently, we apply DPO on the UltraFeedback dataset [25], which emphasizes 3H-values. (c) *Evaluation:* We evaluate our models primarily on three benchmarks: MT-Bench [26], AlpacaEval 2 [27], and Arena-Hard v0.1 [28].

**Results.** AlpacaEval2 and Arena-hard are designed to evaluate the *alignment performance* (3H-values), to assign higher scores to responses preferred by humans [27, 28]. MT-Bench is a benchmark that assesses a model's ability to engage in multi-turn dialogue and accurately *follow instructions* [26]. The results are presented in Table 2, and we have the follow observations:

(a) *Layer Adapter has comparable alignment performance with full fine-tuning.* The SFT model shows strong instruction-following capabilities, as evidenced by its MT-Bench scores. However, its alignment performance reflecting the 3H-value remains limited. However, DPO full fine-tuning significantly enhances alignment performance. Furthermore, we evaluate our proposed InRa with $\lambda = 1.0$. The results indicate that the alignment performance is on par with that of the fully fine-tuned DPO model. Compared to DeRa (which uses SFT and DPO$_{Full}$ models), InRa achieves comparable performance with significantly higher computational efficiency by utilizing only a single adapter layer.

(b) *Interpolation and extrapolation of realignment.* Taking the Arena-hard benchmark as an example, when $\lambda = 0.5$, the alignment strength lies between the SFT model and the InRa model with $\lambda = 1.0$. By appropriately increasing the value of $\lambda$, the alignment ability can be further enhanced, even surpassing the performance of the DPO fully fine-tuned model. A similar phenomenon is observed for AlpacaEval 2 and the first-turn dialogue in MT-Bench.

(c) *InRa offers a quick way to study alignment tax.* However, all MT-Bench results reveal a decline in the model's second-round conversational abilities. As shown in Table 2, the Qwen2.5-7B-Base model's MT-Bench score decreases as $\lambda$ increases in the second-turn dialogue. We attribute this behavior to **alignment tax**. The layer adapter was fine-tuned on the Ultrafeedback dataset, composed solely of single-turn dialogues [25]. As a result, increasing $\lambda$ enhances the model's ability to adhere to single instructions with 3-H values. We provide the case study in Appendix F.2. It also reconfirms the importance of flexible realignment for diverse practical demands.

## 5 In-depth Model Analyses

### 5.1 Layer Significance

We opt to experiment by freezing the lower layers and fine-tuning the top-$k$ layers, as well as by freezing the upper layers and fine-tuning the bottom-$k$ layers. This way, we aim to determine which layers are most effective for alignment. The learning rate is 5e-6, and $\beta$ equals 0.01. As shown in Table 3, tuning the top layers brings limited gains, while adjustments to the bottom layers lead to substantial improvements, highlighting their critical role in preference learning.

Table 3: The significance of layers in alignment on Llama3.1-8B

| Method | AlpacaEval2 | | Arena-Hard | MT-Bench |
| | LC (%) | WR (%) | WR (%) | Score |
| --- | --- | --- | --- | --- |
| top-1 | 4.03 | 4.87 | 8.20 | 5.67 |
| top-3 | 4.05 | 4.96 | 10.40 | 6.32 |
| bottom-1 | 14.94 | 13.93 | 24.80 | 7.37 |
| bottom-3 | **16.38** | **15.51** | **25.30** | **7.40** |

### 5.2 Layer adapter

**Initialzation** For 2D tensors, we use Kaiming initialization [29], and for 1D tensors, we apply standard normalization. As shown in Table 4, a good initialization is more effective when starting from the original weights.

Table 4: Initialization method comparison for layer adapter in alignment on Qwen2.5-7B

| Method | AlpacaEval2 | | Arena-Hard | MT-Bench |
| | LC (%) | WR (%) | WR (%) | Score |
| --- | --- | --- | --- | --- |
| Random | 20.00 | 20.75 | 36.10 | 7.54 |
| Identity copy | **25.74** | **25.83** | **45.10** | **8.21** |

**Comparison with LoRA** We compare our layer adapter with the LoRA method, as shown in Table 5 and Table 6. We fine-tune the LMs using LoRA with ranks $r = 8$ and $r = 128$. Our method achieves performance comparable to full fine-tuning, offering improved training efficiency. Moreover, it demonstrates certain performance advantages over LoRA.

Table 5: Evaluation results across different fine-tuning methods on Qwen2.5-7B.

| Method | AlpacaEval2 | | Arena-Hard | MT-Bench |
| | LC (%) | WR (%) | WR (%) | Score |
| --- | --- | --- | --- | --- |
| Full | 25.20 | 20.92 | **45.50** | 8.21 |
| Lora$_{r=8}$ | 22.04 | 18.64 | 41.50 | 8.16 |
| Lora$_{r=128}$ | 24.97 | 19.22 | 42.50 | 8.42 |
| Ours | **25.74** | **25.83** | 45.10 | 8.21 |

Table 6: Efficiency comparison: training parameters and training time on Qwen2.5-7B.

| Method | Params | Time |
| --- | --- | --- |
| Full | 7264M | 3h35min |
| Lora$_{r=8}$ | 19M | 1h50min |
| Lora$_{r=128}$ | 308M | 1h50min |
| Ours | 222M | 50min |

**Increasing Layer Adapters** In Analysis 5.1, we know that alignment on the bottom layers would be beneficial. Therefore, we are wondering if we can add more adapters preceding the original layers to improve the alignment ability of LM further. We copy the bottom layer $n$ times and perform alignment on these layers. As shown in Table 7,

Table 7: Increasing layer adapters on Llama3.1-8B

| Method | AlpacaEval2 | | Arena-Hard | MT-Bench |
| | LC (%) | WR (%) | WR (%) | Score |
| --- | --- | --- | --- | --- |
| +1 layer | **17.66** | **14.61** | **25.40** | 7.38 |
| +2 layers | 14.27 | 12.67 | 24.70 | **7.42** |
| +3 layers | 14.78 | 13.22 | 23.40 | 7.03 |

increasing the number of layer adapters does not significantly improve performance under the same hyperparameter settings.

**Hyperparameter Stability**
We try different $\beta$ and learning rate combinations in the DPO algorithm to test layer adapter training stability. As shown in Table 8, using a smaller $\beta$ yields significant performance improvements. Moreover, it has been observed that an appropriate $\beta$ value employed in layer adapter training is also well-suited for DPO full fine-tuning. Therefore, our method can be a lightweight proxy for hyperparameter tuning before switching to full fine-tuning. Additional experiments with LoRA, presented in Appendix F.5, further highlight the advantages of our method.

Table 8: Hyperparameter stability on Qwen2.5-7B

| Method | AlpacaEval2 LC (%) | WR (%) | Arena-Hard WR (%) | MT-Bench Score |
|---|---|---|---|---|
| *Layer adapter* | | | | |
| $\beta = 0.01, lr = 5e-6$ | 20.66 | 19.25 | 38.30 | 8.48 |
| $\beta = 0.01, lr = 5e-7$ | 17.53 | 17.34 | 33.20 | 7.87 |
| $\beta = 0.005, lr = 5e-6$ | 19.94 | 19.64 | 39.90 | 7.87 |
| $\beta = 0.005, lr = 5e-7$ | 25.74 | 25.83 | 45.10 | 8.21 |
| *Full* | | | | |
| $\beta = 0.01, lr = 5e-6$ | 12.05 | 12.41 | 31.00 | 7.84 |
| $\beta = 0.01, lr = 5e-7$ | 25.20 | 20.92 | 45.50 | 8.21 |
| $\beta = 0.005, lr = 5e-6$ | 6.09 | 6.83 | 9.70 | 4.56 |
| $\beta = 0.005, lr = 5e-7$ | 26.94 | 22.30 | 46.70 | 8.45 |

# 6  Related Work

**Parameter Efficient Fine Tuning.** Parameter Efficient Fine-Tuning (PEFT) aims to adapt large pre-trained models to downstream tasks by updating only a small subset of parameters, thereby reducing memory and computation costs. Early approaches include adapter modules [30], where small bottleneck layers are inserted into the model and only these are fine-tuned. LoRA [19] introduces low-rank updates to weight matrices, significantly reducing the number of trainable parameters without sacrificing performance. **Our method is orthogonal to these methods.**

**Progressive Learning.** Gong et al. [31] introduced a stacking approach that incrementally doubles model depth to improve training effectiveness. Expanding on this concept, CompoundGrow [32] integrates FeedForward Network expansion into a structured training schedule. More recently, LLama-Pro [33] employs depth growth to retain general model performance while enabling adaptation to domain-specific tasks. **Our work employs depth growth at the lowest layer**.

**Preference Leaning.** RLHF is a method aimed at aligning LLMs with human values and preferences [34]. The PPO algorithm [8] is frequently employed. However, challenges exist throughout the RLHF process, from collecting preference data to training the model, as highlighted by Radford et al. [35]. Alternatively, techniques like DPO [10] eliminate the need for a reward model by training LLMs directly based on human preferences. Other competing methods, including IPO [36], KTO [37], and WSPO [38], have also emerged.

**Realignment.** The most effective way to achieve realignment is by sweeping the hyperparameter. DeRa [18] dynamically adjusts alignment strength at inference time using aligned and unaligned models. Similarly, WSPO [38] demonstrates that when the weak model is identical to the strong model, it can regulate alignment strength during training. We are the first to explore techniques that explicitly incorporate inference-time realignment considerations into the training process.

# 7  Conclusion

We introduce a flexible realignment framework that addresses the realignment of LMs during training and inference. TrRa constructs a controllable teacher signal from existing models, enabling efficient post-training realignment. InRa augments the model with a lightweight layer adapter, supporting inference time alignment adjustment within a single model. Our experiments confirm the practicality of this framework in diverse use cases, such as cost-effective reasoning and dynamic 3H alignment, pointing toward a promising direction for building flexible and user-controllable LLMs.

## Acknowledgments and Disclosure of Funding

Rui is supported by the General Program of the National Natural Science Foundation of China (62176153). Ruobing is supported by the Young Elite Scientists Sponsorship Program by CAST (2023QNRC001).

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

# Appendix

## A    Limitation

For InRa, the inference key-value cache size would double, even though we integrate our new architecture into the vLLM framework [20], which simplifies key-value cache management. However, this may lead to reduced inference throughput. We hope that future work will explore key-value compression techniques to address this issue.

**Future work.**    We list some potential future works as follow:

- **Key-value compression.** We believe that the two key-value cache paths share significant similarities, making key-value compression based on this work a valuable direction for future research.
- **Contrastive Reward Signal.** By injecting short-thinking patterns into the layer adapter and extrapolating between short-thinking and long-thinking logits, the model can be further encouraged to engage in deeper reasoning. Therefore, designing an effective contrastive reward signal could be a promising direction to enhance the model's reasoning capabilities.
- **Realignment.** Current research on training-time realignment remains limited. We hope future work will explore this area more thoroughly, as it holds potential for improving alignment and reasoning performance during model training.
- **Hybrid Model.** A more efficient architecture could support hybrid capabilities, such as dynamically adjustable fast and slow thinking modes, allowing the model to balance speed and reasoning depth based on the task requirements.

## B    Broader Impact

This paper presents work that aims to advance the field of natural language processing. Our work has many potential societal consequences, none of which must be specifically highlighted here.

## C    Proof

### C.1    Approximate Token-Level Distribution

The approximate realigned model $\widehat{\pi}_\theta(\beta/\lambda)$,

$$\pi_\theta^*(\beta/\lambda)(y \mid x) = \frac{\pi^{\mathrm{ref}}(y \mid x) \exp\left[\frac{\lambda}{\beta} r(x, y)\right]}{\sum_{y'} \pi^{\mathrm{ref}}(y' \mid x) \exp\left[\frac{\lambda}{\beta} r(x, y')\right]}. \tag{7}$$

Substituting Equation 3 into Equation 7 yields the following equation:

$$\pi_\theta^*(\beta/\lambda)(y \mid x) = \frac{\pi^{\mathrm{ref}}(y \mid x) \left[\frac{\pi_\theta^*(\beta)(y|x)}{\pi^{\mathrm{ref}}(y|x)}\right]^\lambda}{\sum_{y'} \pi^{\mathrm{ref}}(y' \mid x) \left[\frac{\pi_\theta^*(\beta)(y'|x)}{\pi^{\mathrm{ref}}(y'|x)}\right]^\lambda}. \tag{8}$$

**Proposition 1** *It can be equivalently written as*

$$\widehat{\pi}_\theta(\beta/\lambda)\left(\cdot \mid x, y_{1:t-1}\right) = \mathrm{softmax}\left[\lambda \boldsymbol{h}_t^\theta(\beta) + (1 - \lambda)\boldsymbol{h}_t^{\mathrm{sft}}\right]. \tag{9}$$

***Proof.***    Refer to DeRa paper [18] for the proof.

# D    Justification

**Theorem 1** *As shown in Rafailov et al. [10], any fine-tuned LM $\pi^{\mathrm{ft}}$ and its corresponding pre-trained model $\pi^{\mathrm{ref}}$ can be associated with a reward function $r_{\pi_{\mathrm{ft}}}(x, y)$ such that solving a KL-constrained RL problem yields the fine-tuned model as its optimal policy: $\pi^* \left( r_{\pi_{\mathrm{ft}}}, \pi^{\mathrm{ref}} \right) = \pi^{\mathrm{ft}}$. In particular, the implicit reward can be expressed as $r_{\pi_{\mathrm{ft}}}(x, y) = \beta \log \frac{\pi^{\mathrm{ft}}(y|x)}{\pi^{\mathrm{ref}}(y|x)}$.*

Using Theorem 1, we justify that our experiments in Section 4.2 and Section 4.3 are theoretically well-founded. The models trained via SFT and DPO can be regarded as implicitly learning the reward function embedded in the dataset.

# E    Detailed Experiment

**Model Description**    Deepseek-R1-Distilled-Qwen-1.5B and Deepseek-R1-Distilled-Qwen-7B are fine-tuned using reasoning data generated by DeepSeek-R1 [3]. DeepScaleR-1.5B-Preview, an LM finetuned from Deepseek-R1-Distilled-Qwen-1.5B using simple RL [14].

**Dataset Description**    OpenR1-Math-220k is a large-scale dataset for mathematical reasoning. It consists of 220k math problems with two to four reasoning traces generated by DeepSeek R1 for problems from NuminaMath 1.5. The traces were verified using Math Verify for most samples and Llama-3.3-70B-Instruct as a judge for 12% of the samples, and each problem contains at least one reasoning trace with a correct answer [21].

**Training framework**    The training framework utilizes the LLaMA-Factory [39] repository. All training processes involve full fine-tuning over one epoch with a warm-up ratio of 0.1.

## E.1    Training-time Realignment

We use $\lambda = 1.25$ and $\lambda = 2$ in TrRa to realign the reference model. The loss curves are shown in Figure 5. As observed, the loss converges rapidly, typically within 200 steps. The learning rate is set to $2 \times 10^{-5}$, and the batch size is 16.

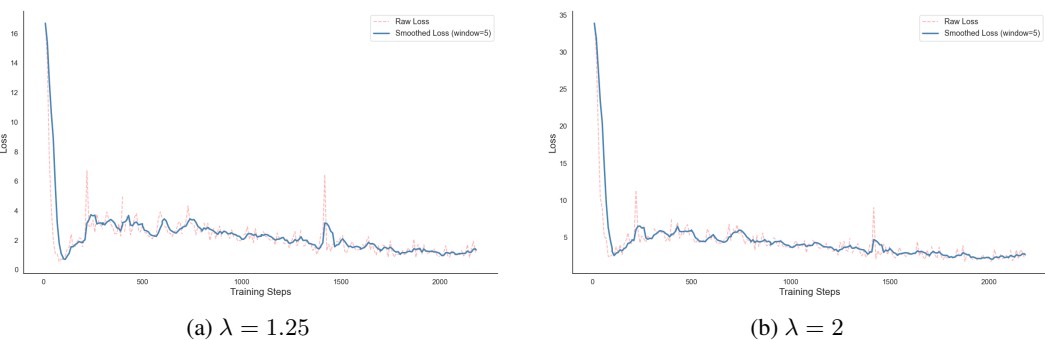

(a) $\lambda = 1.25$          (b) $\lambda = 2$

Figure 5: Comparison of two loss curves.

## E.2    Inference-time Realignment for Reasoning

We employ the short CoT for SFT models, with a learning rate of $2 \times 10^{-5}$ and a batch size of 128. As illustrated in Figure 6, our layer adapter demonstrates its effectiveness—larger models exhibit improved learning performance on the data.

## E.3    Inference-time Realignment for Dialogue Model

**Training details**    The batch size is set to 128. For SFT training, a learning rate of 2e-6 is used for all models. For the DPO-trained model, different learning rates and $\beta$ values are explored, and the best-performing configuration is selected.

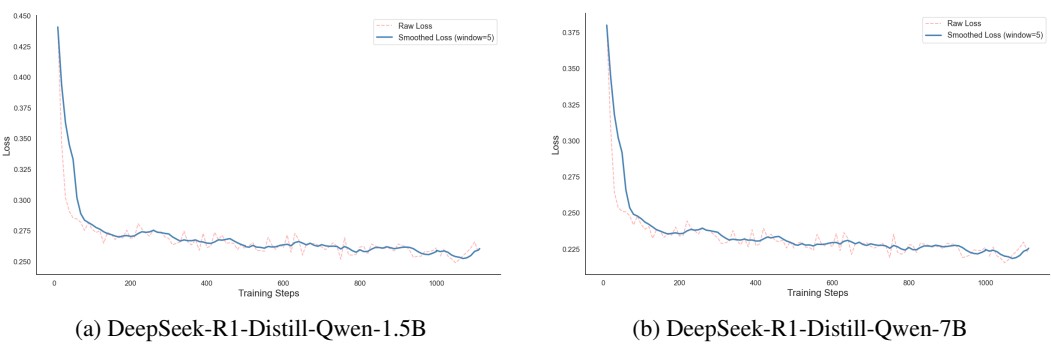

(a) DeepSeek-R1-Distill-Qwen-1.5B          (b) DeepSeek-R1-Distill-Qwen-7B

Figure 6: Comparison of two loss curves.

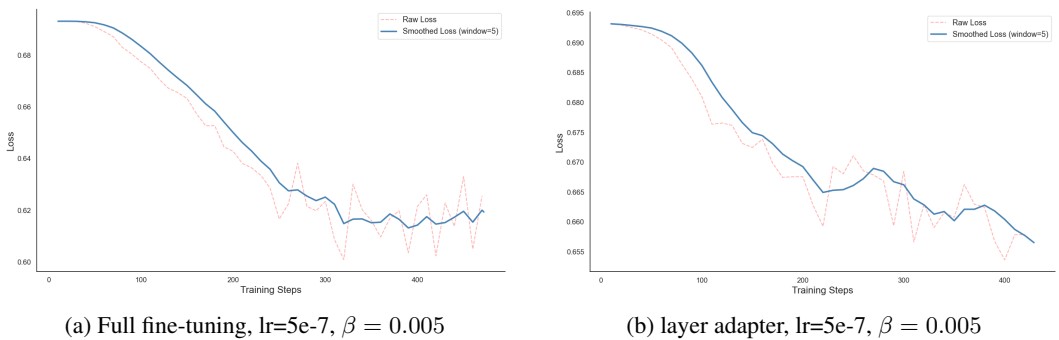

(a) Full fine-tuning, lr=5e-7, $\beta = 0.005$      (b) layer adapter, lr=5e-7, $\beta = 0.005$

Figure 7: Comparison of two loss curves on Qwen2.5-7B model.

As illustrated in Figure 7, these results further demonstrate that, under identical hyperparameter settings, the layer adapter follows a learning trajectory similar to that of full fine-tuning.

**Inference details**     All inferences are conducted using the vLLM engine [20] with a temperature setting of 0.0 (**greedy decoding**) and a maximum generation length of 4096 tokens.

**Judgement**     All these benchmarks are auto-evaluated using LLMs. And we use Qwen2.5-72B-Instruct [23] as the backend API to provide judgment.

### E.4 Numerical Results in Sec4.2

Table 9: Performance comparison of different methods on various benchmarks.

| Models | AIME24 | | AIME25 | | MATH-500 | |
|---|---|---|---|---|---|---|
| | Acc | #Token | Acc | #Token | Acc | #Token |
| DeepSeek-R1-Distill-Qwen-1.5B-InRa ($\lambda = -0.33$) | 32.92 | 12608 | 21.67 | 12732 | 82.88 | 5042 |
| DeepSeek-R1-Distill-Qwen-1.5B-InRa ($\lambda = 0$) | 30.00 | 12602 | 19.58 | 12278 | 80.23 | 4699 |
| DeepSeek-R1-Distill-Qwen-1.5B-InRa ($\lambda = 0.25$) | 25.00 | 11932 | 21.25 | 11368 | 78.65 | 3321 |
| DeepSeek-R1-Distill-Qwen-1.5B-InRa ($\lambda = 0.5$) | 14.17 | 8493 | 12.08 | 8303 | 67.75 | 1765 |
| DeepSeek-R1-Distill-Qwen-1.5B-InRa ($\lambda = 0.75$) | 5.42 | 1211 | 5.42 | 1201 | 61.48 | 515 |
| DeepSeek-R1-Distill-Qwen-1.5B-InRa ($\lambda = 1.0$) | 5.00 | 798 | 4.17 | 633 | 60.75 | 480 |
| DeepSeek-R1-Distill-Qwen-1.5B-InRa ($\lambda = 1.25$) | 3.75 | 661 | 3.33 | 555 | 56.83 | 436 |
| DeepSeek-R1-Distill-Qwen-7B-InRa ($\lambda = -0.25$) | 55.00 | 11224 | 44.17 | 10867 | 88.98 | 3925 |
| DeepSeek-R1-Distill-Qwen-7B-InRa ($\lambda = 0$) | 51.67 | 10576 | 36.67 | 11279 | 87.83 | 3667 |
| DeepSeek-R1-Distill-Qwen-7B-InRa ($\lambda = 0.25$) | 47.50 | 9942 | 31.67 | 9003 | 87.95 | 3073 |
| DeepSeek-R1-Distill-Qwen-7B-InRa ($\lambda = 0.5$) | 42.92 | 8718 | 30.00 | 8629 | 86.40 | 2291 |
| DeepSeek-R1-Distill-Qwen-7B-InRa ($\lambda = 0.75$) | 11.25 | 712 | 7.08 | 608 | 70.85 | 480 |
| DeepSeek-R1-Distill-Qwen-7B-InRa ($\lambda = 1.0$) | 11.25 | 619 | 7.50 | 658 | 70.40 | 440 |
| DeepSeek-R1-Distill-Qwen-7B-InRa ($\lambda = 1.25$) | 7.50 | 616 | 10.83 | 640 | 64.98 | 443 |

# F Supplementary Experiments

## F.1 Alignment Tax

**Alignment Tax**  Alignment tax is the performance incurred to ensure a chatbot's behavior aligns safely and reliably with human values and intentions.

**Experiments**  We conduct reasoning tasks to evaluate whether the layer adapter can learn the reward signal without compromising the foundational capabilities of the LM. We use the zero-shot setting to test the reasoning ability across four benchmarks, including MMLU [40], CMMLU [41], Truthful-QA [42], and GSM8K [43]. We evaluate these benchmarks using *llm-evaluation-harness* [44] repo.

**Results**  As shown in Table 10, reasoning ability decreases slightly as model alignment improves. However, a different trend is observed with TruthfulQA. This is likely because the reward signal incorporates the 3-H values into the model, enhancing its truthfulness.

Table 10: Evaluation results of models across different benchmarks. We evaluate these benchmarks using *llm-evaluation-harness* [44] repo.

| Method | Llama3.1-8B-Base | | | | Qwen2.5-7B-Base | | | |
|---|---|---|---|---|---|---|---|---|
| | MMLU | CMMLU | GSM8K | Truthful-QA | MMLU | CMMLU | GSM8K | Truthful-QA |
| InRa$_{\lambda=0}$ | 59.78 | 47.56 | 57.01 | 54.29 | **71.39** | **81.84** | **82.22** | 56.37 |
| InRa$_{\lambda=0.5}$ | 59.80 | 47.74 | 60.05 | 54.83 | 70.80 | 81.53 | 77.75 | 57.26 |
| InRa$_{\lambda=1.0}$ | 59.91 | 47.86 | **60.65** | 56.90 | 70.37 | 81.28 | 72.71 | 58.44 |
| InRa$_{\lambda=1.5}$ | **60.04** | 47.92 | 57.92 | 59.98 | 70.03 | 80.94 | 68.99 | 59.59 |
| InRa$_{\lambda=2.0}$ | 60.01 | **48.04** | 54.74 | **61.07** | 69.31 | 80.49 | 65.50 | **61.05** |

**Application**  This provides a quick way to identify the alignment tax problem introduced by a specific reward signal.

## F.2 Alignment Tax Verification

As discussed in Section 4.3, alignment extrapolation appears to impair the model's instruction-following capabilities, a phenomenon we term the alignment tax. To substantiate this observation, we conduct the following experiment.

**Experiments**  We perform the following experiments to verify this assumption. We train **one more epoch** during the SFT phase, aiming to reinforce the multi-turn dialogue capability (UltraChat200k is a multi-dialogue dataset).

Table 11: Performance of models on MT-Bench: The SFT model trained for two epochs using the Qwen2-7B-Base model during the SFT phase.

| Method | MT-Bench | |
|---|---|---|
| | 1-turn | 2-turn |
| SFT | 7.60 | 7.20 |
| InRa$_{\lambda=0.5}$ | 8.14 | **7.48** |
| InRa$_{\lambda=1.0}$ | **8.56** | 7.39 |
| InRa$_{\lambda=1.5}$ | 8.48 | 7.21 |
| InRa$_{\lambda=2.0}$ | 8.51 | 7.06 |

**Results**  As shown in Table 11, the performance of the 2-turn dialogue has improved compared to the results presented in Table 2. This observation validates our assumption.

### F.3  Flexible Inference-Time Switching Mechanism

The alignment tax prevents the chatbot from directly following instructions, leading it to prioritize aligning with user preferences—a phenomenon also observed in the GPT-4o incident [13].

**Experiments**  We conduct experiments with varying $\lambda$ values in the multi-turn dialogue phase, utilizing the inference-time realignment capabilities of our InRa.

Table 12: The MT-Bench score on the second turn, using different $\lambda$ values across the two dialogue phases.

| 1-turn / 2-turn | $\lambda = 0.5$ | $\lambda = 1.0$ | $\lambda = 1.5$ | $\lambda = 2.0$ |
|---|---|---|---|---|
| $\lambda = 0.0$ | 7.24 | 7.61 | 7.58 | 7.39 |
| $\lambda = 0.3$ | 7.59 | 7.35 | 7.60 | **7.71** |
| $\lambda = 0.5$ | 7.53 | 7.40 | 7.40 | 7.68 |
| $\lambda = 0.7$ | 7.18 | 7.45 | 7.19 | 7.39 |

**Results**  As shown in Table 12, our findings indicate that lowering the $\lambda$ value in the second-turn dialogue helps the model better comprehend the context and follow instructions more effectively, **avoiding the alignment tax problem.**  Refer to Appendix F.6 for a detailed case study.

### F.4  The Function of Layer Adapter

We provide a visualization to illustrate the function of the layer adapter. As shown in Figure 8, the adapter does not project the original input embeddings into a higher-dimensional space; instead, it maps the unaligned input embeddings to their aligned counterparts.

### F.5  The results of Lora

**Experiments**  We sweep the hyperparameter to test the different methods.

**Results**  As shown in Table 13, LoRA appears to require a high learning rate, whereas full fine-tuning demands a lower learning rate to maintain language ability and achieve strong performance. Our method aligns closely with full fine-tuning and remains stable throughout different hyperparameters.

We further increased the learning rate to train LoRA, and the results are presented in Table 14. As shown, increasing the learning rate degrades the model's language capabilities. Therefore, these configurations did not produce optimal results.

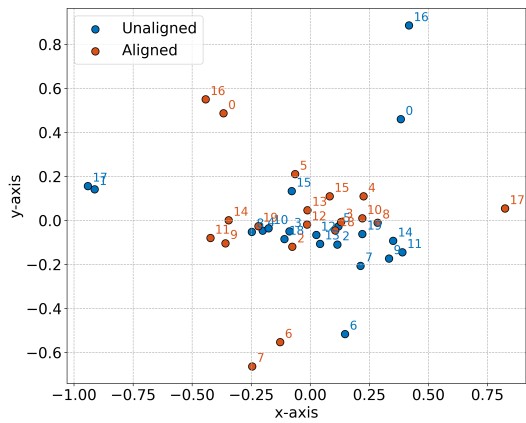

Figure 8: Visualization of input embeddings using the PCA method, with the ID numbers representing token positions.

Table 13: Hyperparameter stability on Qwen2.5-7B

| Method | AlpacaEval2 LC (%) | WR (%) | Arena-Hard WR (%) | MT-Bench Score |
|---|---|---|---|---|
| *Layer adapter* | | | | |
| $\beta = 0.01, lr = 5e-6$ | 20.66 | 19.25 | 38.30 | 8.48 |
| $\beta = 0.01, lr = 5e-7$ | 17.53 | 17.34 | 33.20 | 7.87 |
| $\beta = 0.005, lr = 5e-6$ | 19.94 | 19.64 | 39.90 | 7.87 |
| $\beta = 0.005, lr = 5e-7$ | 25.74 | 25.83 | 45.10 | 8.21 |
| *LoRA, $r = 8$* | | | | |
| $\beta = 0.01, lr = 5e-6$ | 22.04 | 18.64 | 41.50 | 8.16 |
| $\beta = 0.01, lr = 5e-7$ | 4.33 | 2.84 | 11.30 | 7.74 |
| $\beta = 0.005, lr = 5e-6$ | 17.99 | 14.98 | 36.10 | 7.99 |
| $\beta = 0.005, lr = 5e-7$ | 5.31 | 3.24 | 11.90 | 7.75 |
| *LoRA, $r = 128$* | | | | |
| $\beta = 0.01, lr = 5e-6$ | 21.50 | 17.56 | 37.30 | 8.33 |
| $\beta = 0.01, lr = 5e-7$ | 6.29 | 4.25 | 13.50 | 7.68 |
| $\beta = 0.005, lr = 5e-6$ | 24.97 | 19.22 | 42.50 | 8.42 |
| $\beta = 0.005, lr = 5e-7$ | 7.12 | 4.46 | 13.50 | 7.64 |
| *Full* | | | | |
| $\beta = 0.01, lr = 5e-6$ | 12.05 | 12.41 | 31.00 | 7.84 |
| $\beta = 0.01, lr = 5e-7$ | 25.20 | 20.92 | 45.50 | 8.21 |
| $\beta = 0.005, lr = 5e-6$ | 6.09 | 6.83 | 9.70 | 4.56 |
| $\beta = 0.005, lr = 5e-7$ | 26.94 | 22.30 | 46.70 | 8.45 |

Table 14: Increasing layer adapters on Qwen2.5-7B

| Method | AlpacaEval2 LC (%) | WR (%) | Arena-Hard WR (%) |
|---|---|---|---|
| $\beta = 0.01, lr = 2e-5$ | 16.68 | 16.89 | 34.90 |
| $\beta = 0.01, lr = 5e-5$ | 14.06 | 14.83 | 24.20 |
| $\beta = 0.01, lr = 1e-4$ | 12.26 | 12.51 | 19.80 |
| $\beta = 0.005, lr = 2e-5$ | 0.42 | 1.05 | 0.70 |
| $\beta = 0.005, lr = 5e-5$ | 4.69 | 4.34 | 2.10 |
| $\beta = 0.005, lr = 1e-4$ | 0.16 | 0.49 | 1.80 |

**Conclusion**    Our method can be a lightweight proxy for hyperparameter tuning before switching to full fine-tuning. This is possible because the optimal hyperparameters tend to be relatively consistent between the two.

## F.6    Alignment Tax Case Study

This section presents a case from MT-Bench that highlights how the alignment tax, introduced by catering to human preferences, can impair an LM's ability to manage multi-turn dialogues.

**1. We begin by presenting the first-turn instruction as follows:**

---
**1-turn Insturction**

Read the below passage carefully and answer the questions with an explanation: At a small company, parking spaces are reserved for the top executives: CEO, president, vice president, secretary, and treasurer with the spaces lined up in that order. The parking lot guard can tell at a glance if the cars are parked correctly by looking at the color of the cars. The cars are yellow, green, purple, red, and blue, and the executives' names are Alice, Bert, Cheryl, David, and Enid.
* The car in the first space is red.
*A blue car is parked between the red car and the green car.
* The car in the last space is purple.
* The secretary drives a yellow car.
* Alice's car is parked next to David's.
* Enid drives a green car.
* Bert's car is parked between Cheryl's and Enid's.
* David's car is parked in the last space.
Question: What is the name of the secretary?

---

**2. The model can follow the first-turn instruction well.**

---
**1-turn Response**

The secretary's car is identified by two clues in the passage:
1. The secretary drives a yellow car.
2. Alice's car is parked next to David's.
Since David's car is parked in the last space (the fifth position), and the car in the last space is purple, David cannot be the secretary. Therefore, Alice must be the secretary because her car is next to David's, and the only car left that matches the secretary's description is yellow.
The name of the secretary is Alice. ✓

---

**3. We go on to request the second instruction.**

---
**2-turn Instruction**

List car colors in order from last to first.

---

**4. However, the model follows the second instruction well but fails to incorporate the context when generating its response.**

**5. We modify the second-turn instruction to encourage the model to use context when answering questions.**

**2-turn Instruction**

According to the above passage, list car colors in order from last to first.

**6. As shown below, the model successfully provides the correct answer.**

**2-turn Response**

The passage does not provide a list of car colors in order from last to first. Instead, it describes specific car locations and colors based on certain positions. To correct the order, we need additional information. Here is the information given in the passage:
1. The car in the first space is red.
2. A blue car is parked between the red car and the green car.
3. The car in the last space is purple.
4. The secretary drives a yellow car.
5. Alice's car is parked next to David's.
6. Enid drives a green car.
7. Bert's car is parked between Cheryl's and Enid's.
8. David's car is parked in the last space.
From this, we can deduce the following order based on the positions:
1. Red car (first space)
2. Blue car (second space)
3. Green car (third space)
4. Yellow car (last before the last space)
5. Purple car (last space)
So, the car colors in order from last to first as described in the passage are: Purple, Yellow, Green, Blue, Red. ✓

## F.7 Thinking Case Study

**Problem**

Find the sum of all integer bases $b > 9$ for which $17_b$ is a divisor of $97_b$.

---

**DeepSeek-R1-Distill-Qwen-7B-InRa $\lambda$ : 1.0 tokens: 429**

To find the sum of all integer bases $b > 9$ for which $17_b$ is a divisor of $97_b$, we start by converting the numbers from base $b$ to base 10.
- $17_b$ in base 10 is $1 \cdot b + 7 = b + 7$. - $97_b$ in base 10 is $9 \cdot b + 7 = 9b + 7$.
We need $17_b$ to divide $97_b$, which translates to finding $b > 9$ such that $b + 7$ divides $9b + 7$.
This can be expressed as:
$$\frac{9b + 7}{b + 7} \text{ is an integer}$$
To solve this, we use polynomial division or the Euclidean algorithm. Dividing $9b + 7$ by $b + 7$ gives:
$$9b + 7 = 9(b + 7) - 65$$
Thus, the remainder is $-65$. For $b + 7$ to divide $9b + 7$, it must divide 65. The divisors of 65 are 1, 5, 13, and 65. Since $b > 9$, we consider only the divisors 13 and 65, leading to: - $b + 7 = 13$ which gives $b = 6$ (not valid as $b > 9$) - $b + 7 = 65$ which gives $b = 58$
Thus, the only valid base is $b = 58$. The sum of all such bases is simply 58.

$$\boxed{58}$$

---

**DeepSeek-R1-Distill-Qwen-7B-InRa $\lambda$ : 0.6 tokens: 720**

<think> Alright, let me try to solve this problem step by step. So, the question is: Find the sum of all integer bases $b > 9$ for which $17_b$ is a divisor of $97_b$.
First, I need to understand what $17_b$ and $97_b$ mean in base $b$. In any base $b$, a number like $17_b$ translates to $1 \times b + 7$ in decimal, right? Similarly, $97_b$ translates to $9 \times b + 7$. So, I need to express these values in decimal form.
Calculating the decimal equivalents: - $17_b = 1 \times b + 7 = b + 7$ - $97_b = 9 \times b + 7 = 9b + 7$
Now, the problem states that $17_b$ is a divisor of $97_b$. This translates to the condition that $b + 7$ must divide $9b + 7$.
Mathematically, this means:
$$9b + 7 \equiv 0 \pmod{b + 7}$$
I remember that if $a$ divides $b$, then $b \mod a = 0$. So, applying this here:
$$9b + 7 \equiv 0 \pmod{b + 7}$$
To solve this congruence, I can use the property that $b \equiv -7 \pmod{b + 7}$. This is because $b = (b + 7) - 7$, so $b \equiv -7 \pmod{b + 7}$.
Substituting $b \equiv -7$ into $9b + 7$:
$$9(-7) + 7 = -63 + 7 = -56$$
Thus, the condition simplifies to:
$$-56 \equiv 0 \pmod{b + 7}$$
$b + 7$ must be a divisor of -56.
Since $b + 7$ is a positive integer (as $b > 9$), it must be a positive divisor of 56. The divisors of 56 are:
$$1, 2, 4, 7, 8, 14, 28, 56$$
Since $b + 7 > 16$ (because $b > 9$), the possible values of $b + 7$ are 28 and 56.
Therefore, solving for $b$: - If $b + 7 = 28$, then $b = 21$ - If $b + 7 = 56$, then $b = 49$
Both bases 21 and 49 are greater than 9. The problem asks for the sum of all such bases.
Thus, the sum is $21 + 49 = 70$.
**Final Answer** The sum of all integer bases $b > 9$ is $\boxed{70}$.

<think> Alright, so I have this problem here: I need to find the sum of all integer bases $b > 9$ for which $17_b$ is a divisor of $97_b$. Hmm, okay. Let me try to break this down step by step.

First, I know that numbers in a base $b$ can be converted to decimal (base 10) by multiplying each digit by $b$ raised to the power of its position, starting from 0 on the right. So, let me convert both numbers $17_b$ and $97_b$ into decimal.

Starting with $17_b$: - The rightmost digit is 7, which is in the $b^0$ place, so that's $7 \times 1 = 7$. - The next digit to the left is 1, which is in the $b^1$ place, so that's $1 \times b = b$. - Adding them together, $17_b$ in decimal is $b + 7$.

Now, $97_b$: - The rightmost digit is 7, so that's $7 \times 1 = 7$. - The next digit is 9, which is in the $b^1$ place, so that's $9 \times b = 9b$. - Adding them together, $97_b$ in decimal is $9b + 7$.

So, the problem simplifies to finding all integer bases $b > 9$ where $b + 7$ divides $9b + 7$. Let's denote $d = b + 7$. Then, $d$ must divide $9b + 7$.

To find this divisibility, let me express $9b + 7$ in terms of $d$. Since $d = b + 7$, we can solve for $b$:

$$b = d - 7$$

Substituting $b$ into $9b + 7$:

$$9(d - 7) + 7 = 9d - 63 + 7 = 9d - 56$$

So, $d$ must divide $9d - 56$. This implies: $d$ divides $9d - 56$

Subtracting $9d$ from both sides, we get: $d$ divides $-56$

Since divisibility is unaffected by the sign, $d$ must be a positive divisor of 56. The positive divisors of 56 are:

$$1, 2, 4, 7, 8, 14, 28, 56$$

But $d = b + 7$ and $b > 9$, so $d$ must be greater than $9 + 7 = 16$. Looking at the list of divisors, the ones greater than 16 are 28 and 56. Therefore, the possible values for $d$ are 28 and 56.

Calculating $b$ for each: - If $d = 28$, then $b = 28 - 7 = 21$ - If $d = 56$, then $b = 56 - 7 = 49$

So, the bases $b$ are 21 and 49. Adding them together:

$$21 + 49 = 70$$

Therefore, the sum of all integer bases $b > 9$ for which $17_b$ is a divisor of $97_b$ is $\boxed{70}$.

**Wait a second...** Let me double-check my reasoning. Could there be any other divisors I missed?

The divisors of 56 are indeed $1, 2, 4, 7, 8, 14, 28, 56$. Those greater than 16 are only 28 and 56. Since $d$ must be greater than 16, 21 and 49 are the only valid bases. Yep, that seems right.

Just to be thorough, let me verify with these bases:

1. For $b = 21$: - $17_{21} = 21 + 7 = 28$ - $97_{21} = 9 \times 21 + 7 = 189 + 7 = 196$ - Checking if 28 divides 196: $196 28 = 7$, which is an integer. So, yes, 28 divides 196.

2. For $b = 49$: - $17_{49} = 49 + 7 = 56$ - $97_{49} = 9 \times 49 + 7 = 441 + 7 = 448$ - Checking if 56 divides 448: $448 56 = 8$, which is an integer. So, yes, 56 divides 448.

All checks out. So, the answer must be correct.

**Final Answer** $\boxed{70}$ </think>

To find the sum of all integer bases $b > 9$ for which $17_b$ is a divisor of $97_b$, we first convert the numbers from base $b$ to decimal.

- The number $17_b$ in decimal is $1 \times b + 7 = b + 7$. - The number $97_b$ in decimal is $9 \times b + 7 = 9b + 7$.

We need $b + 7$ to be a divisor of $9b + 7$. Let $d = b + 7$. Then, $d$ must divide $9b + 7$. Substituting $b = d - 7$ into $9b + 7$ gives:

$$9(d - 7) + 7 = 9d - 56$$

Thus, $d$ must divide $9d - 56$, which implies $d$ divides $-56$. Therefore, $d$ must be a positive divisor of 56. The positive divisors of 56 are:

$$1, 2, 4, 7, 8, 14, 28, 56$$

Since $d = b + 7$ and $b > 9$, $d$ must be greater than 16. The valid divisors are 28 and 56. Calculating $b$ for each: - If $d = 28$, then $b = 28 - 7 = 21$ - If $d = 56$, then $b = 56 - 7 = 49$

Thus, the bases $b$ are 21 and 49. Adding them together:

$$21 + 49 = 70$$

Therefore, the sum of all integer bases $b > 9$ for which $17_b$ is a divisor of $97_b$ is $\boxed{70}$.

<think> Alright, so I have this problem here: I need to find the sum of all integer bases $b > 9$ for which $17_b$ is a divisor of $97_b$. Hmm, okay. Let me try to break this down step by step.

First, let me recall what it means for a number to be in a certain base. If I have a number like $17_b$, that means in base $b$, it's equal to $1 \times b + 7 \times 1$ in decimal, right? Similarly, $97_b$ would be $9 \times b + 7 \times 1$ in decimal. So, in math terms, $17_b = 1 \times b + 7 = b + 7$ and $97_b = 9 \times b + 7 = 9b + 7$.

Now, the problem says that $17_b$ is a divisor of $97_b$. So, in decimal terms, that means $b + 7$ divides $9b + 7$. In other words, $b + 7$ is a factor of $9b + 7$. How can I translate this into an equation?

Well, if $b + 7$ divides $9b + 7$, then $9b + 7$ divided by $b + 7$ should give an integer. Maybe I can perform the division and see what that gives me.

Let me set up the division: $\frac{9b+7}{b+7}$. Let's do polynomial division or maybe use algebra to simplify this.

Let's express $9b + 7$ in terms of $b + 7$. Notice that $9b + 7 = 9(b+7) - 63 + 7 = 9(b+7) - 56$. Wait, let me check that. If I expand $9(b + 7)$, it's $9b + 63$. Then subtracting 56 gives $9b + 7$. Yeah, that works.

So, $9b + 7 = 9(b + 7) - 56$. Therefore, dividing by $b + 7$, we get $\frac{9b+7}{b+7} = 9 - \frac{56}{b+7}$.

For this to be an integer, $\frac{56}{b+7}$ must be an integer. So, $b + 7$ must be a divisor of 56. Got it.

So, $b + 7$ divides 56. Therefore, $b + 7$ must be one of the positive divisors of 56. Since $b$ is an integer greater than 9, let's find all positive divisors of 56, subtract 7, and see which of those results are greater than 9.

First, let's list all positive divisors of 56. The divisors of 56 are: 1, 2, 4, 7, 8, 14, 28, 56.

Now, subtracting 7 from each of these, we get:

1 - 7 = -6 2 - 7 = -5 4 - 7 = -3 7 - 7 = 0 8 - 7 = 1 14 - 7 = 7 28 - 7 = 21 56 - 7 = 49

So, the possible values for $b$ are these results, but since $b$ has to be an integer greater than 9, we can ignore the negative numbers and zero. That leaves us with 1, 7, 21, 49. Wait, but 1, 7 are not greater than 9. So, only 21 and 49 are valid.

Wait, hold on, are these correct? Let me verify.

If $b + 7$ is a divisor of 56, then $b + 7$ can be 8, 14, 28, 56. Since $b > 9$, $b + 7 > 16$. So, the possible $b + 7$ must be divisors of 56 that are greater than 16.

Looking back at the list of divisors: 56's divisors are 1, 2, 4, 7, 8, 14, 28, 56.

Divisors greater than 16 are 28 and 56.

So, $b + 7 = 28$ implies $b = 28 - 7 = 21$.

$b + 7 = 56$ implies $b = 56 - 7 = 49$.

Ah, so I was wrong earlier. I should have considered only the $b + 7$ greater than 16, which gives $b$ as 21 and 49.

Therefore, the integer bases $b$ greater than 9 are 21 and 49.

Therefore, the sum of these is $21 + 49 = 70$.

Wait, just to make sure I haven't missed anything.

Let me double-check:

Compute $17_{21}$ and $97_{21}$ in decimal:

$17_{21} = 1 \times 21 + 7 = 28$

$97_{21} = 9 \times 21 + 7 = 189 + 7 = 196$

Is 28 a divisor of 196? Yes, because 196 / 28 = 7, which is an integer.

Similarly, for base 49:

$17_{49} = 1 \times 49 + 7 = 56$

$97_{49} = 9 \times 49 + 7 = 441 + 7 = 448$

Is 56 a divisor of 448? 448 / 56 = 8, which is an integer. Perfect.

Wait a second, just to make sure, are there any other divisors between, say, 16 and 56? Let's see:

Divisors of 56 are as listed: 1, 2, 4, 7, 8, 14, 28, 56. So, between 16 and 56, the only one is 28 and 56.

Wait, actually 56 is included in the first step when I subtracted 7, but 56 is just one of the divisors.

So, so I think I got all possibilities.

Wait, but let's make sure that $b$ must be greater than 9, so in base $b$, the digits must be less than $b$. In $97_b$, the digit 9 must be less than $b$, which means $b > 9$. So, that is already taken care of.

Therefore, the bases $b > 9$ are 21 and 49, and their sum is 70.

Therefore, the final answer is 70.

**Final Answer** $\boxed{70}$ </think>

To solve the problem, we need to find the sum of all integer bases $b > 9$ for which $17_b$ is a divisor of $97_b$.

First, we convert the numbers from base $b$ to decimal: - $17_b = 1 \times b + 7 = b + 7$ - $97_b = 9 \times b + 7 = 9b + 7$

We need $b + 7$ to be a divisor of $9b + 7$. This can be expressed as:

$$\frac{9b + 7}{b + 7}$$

We simplify this division by expressing $9b + 7$ in terms of $b + 7$:

$$9b + 7 = 9(b + 7) - 56$$

Thus, the division becomes:

$$\frac{9b + 7}{b + 7} = 9 - \frac{56}{b + 7}$$

For this to be an integer, $\frac{56}{b+7}$ must be an integer. Therefore, $b + 7$ must be a divisor of 56.

The divisors of 56 are: 1, 2, 4, 7, 8, 14, 28, 56.

Since $b > 9$, we need $b + 7$ to be a divisor of 56 greater than 16. The valid divisors are 28 and 56: - If $b + 7 = 28$, then $b = 21$ - If $b + 7 = 56$, then $b = 49$

Thus, the valid bases are 21 and 49. Their sum is:

$$21 + 49 = 70$$

Therefore, the final answer is:

$$\boxed{70}$$

---

### DeepSeek-R1-Distill-Qwen-7B-InRa $\lambda : -0.5$ tokens: 3380

<think> Alright, so I have this problem here: I need to find the sum of all integer bases $b > 9$ for which $17_b$ is a divisor of $97_b$. Hmm, okay, let me try to parse this out step by step.

First off, when numbers are written with a subscript like $17_b$ or $97_b$, that means they're numbers in base $b$, right? So $17_b$ isn't one seven in base b, but rather, it's the number "one" and "seven" in base b. Similarly, $97_b$ is nine and seven in base b. Okay, got it.

So, in base 10, we're used to numbers like 17 being "one ten and seven units," which is 1*10 + 7*1. Similarly, 97 in base 10 is nine tens and seven units. But since we're dealing with an unknown base $b$, I think we need to convert these numbers into base 10 to work with them more easily, since divisibility is something we can handle in base 10.

Alright, so let's try to convert $17_b$ and $97_b$ into base 10. In base $b$, the rightmost digit is the units place, which is $b^0$, then the next is the $b^1$ place, and so on.

So for $17_b$: The rightmost digit is 7, which is 7 in base 10, and the next digit is 1, which is $1 \times b^1$. So adding those together, $17_b$ in base 10 is $1 \times b + 7 \times 1 = b + 7$. Got that.

Similarly, for $97_b$: The rightmost digit is 7, so that's $7 \times 1$, and the next digit is 9, which is $9 \times b$. Therefore, $97_b$ in base 10 is $9 \times b + 7 \times 1 = 9b + 7$.

So now we have $17_b = b + 7$ and $97_b = 9b + 7$.

The problem states that $17_b$ is a divisor of $97_b$, which in other words, $97_b$ divided by $17_b$ is an integer. So mathematically, that means $(9b + 7)$ divided by $(b + 7)$ is an integer. So, written as a fraction, it would be $\frac{9b+7}{b+7}$, and we want this to be an integer. Let me note that down:

$\frac{9b+7}{b+7}$ must be an integer.

Our goal is to find all integer bases $b > 9$ for which this fraction is an integer, and then sum those $b$s.

So, to approach this, maybe I can perform the division of $9b+7$ by $b+7$ and see if it simplifies in a way that gives us some equation we can work with.

Let me set $N = 9b + 7$ and $D = b + 7$, so we have $N/D$ must be integer.

Let me try to perform the division.

Divide N by D: $(9b + 7) \div (b + 7)$.

Let me do polynomial long division or just think about how many times $b + 7$ goes into $9b + 7$.

Well, $b + 7$ multiplied by 9 is $9b + 63$. Hmm, let's subtract that from $9b + 7$:

$9b + 7 - (9b + 63) = 0b - 56$. So, we have a remainder of $-56$. So, in other words: $\frac{9b+7}{b+7} = 9 - \frac{56}{b+7}$.

So, for $\frac{9b+7}{b+7}$ to be an integer, that implies that $\frac{56}{b+7}$ must also be an integer because 9 is already an integer, so their difference has to be an integer. So, $\frac{56}{b+7}$ must be an integer.

So, $\frac{56}{b+7}$ is integer. Therefore, $b + 7$ must be a divisor of 56. Since $b > 9$, $b + 7 > 9 + 7 = 16$.

So, $b + 7$ must be a positive divisor of 56 that's greater than 16.

Therefore, we can find all the positive divisors of 56, then exclude those that are 16 or less, and the remaining ones would correspond to possible values of $b + 7$, hence we can find $b$ by subtracting 7.

Okay, so first, let me list all positive divisors of 56.

56 is equal to $7 \times 8$, which is $7 \times 2^3$, so its prime factors are 2, 2, 2, 7. So the divisors can be found by taking combinations of these exponents.

The divisors of 56 are: 1, 2, 4, 7, 8, 14, 28, 56.

Wait, let me check that: starting from 1, since 1 divides everything.

Then, 2 because 56 is even.

4 because 56 ÷ 2 is 28, which is 2×14, then 7 is a prime, so 7 divides 56 since 7×8=56.

8 is 8×7, so 8 is a divisor. Then, 14 is 14×4=56? Wait, 14×4 is 56, yes. 14×4=56.

Then, 28 is 28×2=56, and 56 itself is a divisor.

So the positive divisors are 1, 2, 4, 7, 8, 14, 28, 56. So, that is eight divisors.

Now, since $b > 9$, $b + 7 > 16$. So, we need divisors of 56 that are greater than 16.

Looking at the list: 1, 2, 4, 7, 8, 14, 28, 56.

Which of these are greater than 16?

28, and 56. 14 is less than 16, so 28 and 56.

Wait, but 14 is 14, which is less than 16, so doesn't qualify.

Therefore, the only divisors greater than 16 are 28 and 56.

Thus, $b + 7$ can be either 28 or 56, so $b$ can be 28 - 7 = 21, or 56 - 7 = 49.

Wait, that seems straightforward, but let me check if that's right.

So, $b + 7 = 28 \implies b = 21$, and $b + 7 = 56 \implies b = 49$.

Is that all? Let me make sure.

Wait, hold on a second. The problem mentions that in base $b$, the digits in $17_b$ and $97_b$ have digits 1, 7 and 9, 7. So, digits in base $b$ have to be less than the base.

Therefore, in $17_b$, the digits are 1 and 7, so 7 has to be less than $b$. Similarly, in $97_b$, the digits are 9 and 7, so 9 has to be less than $b$.

Therefore, $b$ must be greater than 9. Wait, the problem already states $b > 9$. So, 7 is less than $b$, so no additional constraints from that. So, that just reiterates that $b > 9$.

Therefore, our only possible bases are 21 and 49.

Wait, but let me make sure that when we have these bases, the initial numbers make sense in that base. For $17_b$, digits 1 and 7, which are both less than 21 or 49, so that's fine. Similarly, $97_b$ has digit 9, which is less than both 21 and 49, so that's fine too.

Therefore, seems like b = 21 and b = 49 satisfy the condition.

Hence, the sum is 21 + 49.

Calculating 21 + 49: 20 + 40 = 60, 1 + 9 = 10, so 70.

So, the sum should be 70.

But let me just double-check to make sure I haven't made any mistakes.

First, let's check for b = 21.

Calculating $17_{21}$ is $1 * 21 + 7 = 28$.

Calculating $97_{21}$ is $9 * 21 + 7$. 9*20 is 180, plus 9 is 189, plus 7 is 196.

So, 196 divided by 28 is equal to 7, which is an integer. So, yes, 21 is valid.

Now, let's check b = 49.

Calculating $17_{49}$: $1 * 49 + 7 = 56$.

Calculating $97_{49}$: $9 * 49 + 7$. 9*40 is 360, 9*9 is 81, so 360 + 81 = 441, plus 7 is 448.

Dividing 448 by 56. 56*8 is 448, so that's 8, which is an integer. So, yes, 49 is also valid.

So, indeed, both 21 and 49 are valid bases, and their sum is 70.

Therefore, I think 70 is the answer.

But let me just go back and check if there could be other divisors which I may have missed. Because sometimes, when we're working with divisors, we might miss some, especially if we're only considering positive integers.

Wait, let me list all the positive divisors of 56 again. 1, 2, 4, 7, 8, 14, 28, 56. So that is all.

Are there any negative divisors? Well, technically, yes, 56 has also negative divisors: -1, -2, -4, -7, -8, -14, -28, -56.

But since $b$ is a base, and base cannot be negative, so we only consider positive divisors.

Additionally, $b + 7$ is a positive number greater than 16.

Therefore, we only considered 28 and 56, corresponding to $b = 21$ and $b = 49$. So that seems complete.

I was just thinking, sometimes with bases, the digits have to satisfy that all digits are less than the base. So, in this case, our digits are 1,7 and 9. Therefore, for each number, in its respective base, all digits must be less than the base.

But as I mentioned earlier, for $17_b$, digits are 1 and 7, so base must be greater than 7, and for $97_b$, digits 9 and 7, so base must be greater than 9.

Since the problem already specifies $b > 9$, so in our case, $b = 21$ and 49 both satisfy that, so no issues there.

I also double-checked the division earlier on, seems to hold, so my conclusion seems solid.

Therefore, I'm confident that the answer is 21 + 49 = 70.

**Final Answer** The sum of all integer bases $b > 9$ is $\boxed{70}$. </think>

To solve the problem, we need to find the sum of all integer bases $b > 9$ for which $17_b$ is a divisor of $97_b$.

First, we convert the numbers from base $b$ to base 10: - $17_b$ in base 10 is $1 \times b + 7 = b + 7$. - $97_b$ in base 10 is $9 \times b + 7 = 9b + 7$.

We need $17_b$ to be a divisor of $97_b$, which means $\frac{9b+7}{b+7}$ must be an integer. Performing polynomial long division, we get:

$$\frac{9b + 7}{b + 7} = 9 - \frac{56}{b + 7}$$

For this to be an integer, $\frac{56}{b+7}$ must be an integer. Therefore, $b + 7$ must be a divisor of 56.

The divisors of 56 are: 1, 2, 4, 7, 8, 14, 28, 56. Since $b > 9$, $b + 7$ must be greater than 16. The suitable divisors are 28 and 56.

Thus, $b + 7 = 28$ gives $b = 21$ and $b + 7 = 56$ gives $b = 49$.

The sum of these bases is $21 + 49 = 70$.

The sum of all integer bases $b > 9$ is $\boxed{70}$.

