# OpenReview forum: "Flexible Realignment of Language Models"
_NeurIPS.cc/2025/Conference — NeurIPS 2025 poster_

### Official Review · Reviewer_PCc3 · 2025-06-30

**Clarity:** 3
**Significance:** 3
**Originality:** 3
**Rating:** 5
**Confidence:** 3

**Summary:**

This paper proposes a flexible realignment framework for LLMs, including training-time realignment and inference-time realignment. Training-time realignment utilizes logits, and inference-time realignment is implemented with a layer adaptor. Experiments on math reasoning and dialogue models show effectiveness.

**Questions:**

1. When saying Llama 3.1 8B under the dialogue model, are you referring to chat models?
2. I wonder if you did any ablation study on the impact of adapting different layer depths?

**Ethical Concerns:**

["NO or VERY MINOR ethics concerns only"]

**Final Justification:**

The author has addressed my concerns. I have increased my score.

**Limitations:**

Yes, the author addressed the limitations in Appendix

**Quality:**

3

**Strengths And Weaknesses:**

## Strength
1. Their framework includes LM realignment during both training and inference time. The framework is quite flexible and dynamic.
2. They study various models. For training and inference realignment, they study Deepseek. They expand their experiment to dialogue models, including Llama and Qwen.

## Weakness
1. One concern is that the model Inference-time realignment performance relies a lot on the hyperparameter λ. The optimal λ varies across the tasks. It would be improved if the authors could provide more discussion on the intuition for how to tune λ.
2. TrRa-iter sacrifices correctness for token reduction, as well as in some cases of no iterations. It would be better if the authors could provide more discussion.
3. It would be better if they could discuss more about the motivation for studying dialogue models and the training-time realignment of those.

---

> ### Author Rebuttal · Authors · 2025-07-30
>
> ```
>  |-----------|-----------|-------------------|-----------|------>
>  0.0         0.5        1.0                 2~5          >5.0
>  (Base)       ↓      (already aligned)   (More aligned)  ⚠️ Reward Hacking Risk
>               |
>         Base → Aligned
> ```
> **W1: Intutition of $\lambda$ in InRa**
> > - The origin is the base model, and the aligned model lies to the right. If alignment feels too strong, set $\lambda$ between 0 and 1; if it's too weak, set it above 1. Values around 2–5 are generally safe; beyond that may cause reward hacking.
> >
> > - It is implemented as a single API parameter—similar to `temperature` and `top_p  `(lower for more deterministic, higher for more diverse)—that you can flexibly adjust according to your needs.
> > - For instance, as shown in experiments in Section 4.2, $\lambda$ enables adaptive reasoning **based on the user's budget.** The token reduction performance across three benchmarks (blue curve) is consistent.
>
> **W2: `TrRa-iter sacrifices correctness for token reduction, as well as in some cases of no iterations.`**
>
> > - In Table 1 of no iteration, only the case of $\lambda = 10$ shows a 0.42% performance drop on AIME-24, which may be attributed to randomness. In contrast, on AIME-25 and Math-500, the performance consistently improves by over 3% on average.
> > - **The table sweeps $\lambda$ values, which are more intended to reflect the flexibility and correctness of TrRa.**
> >
> > - **TrRa-iter** further validates our algorithm and demonstrates that the reward signal can be **effectively transferred**. However, performing TrRa-iter may not be strictly necessary in practical applications.
> >   - Suppose you use λ=2 in iteration 1, resulting in a better-aligned model.
> >   - If you then use λ=2 again in iteration 2, it may be equivalent to λ>4 in iteration 1 (based on our empirical experiments).
> >   - Therefore, due to **reward hacking**, performance degradation becomes **unavoidable** with more iterations of TrRa.
> >  - OpenAI o3 and DeepSeek R1 also suggest that the **short chain of thought may not benefit model reasoning.**
>
> **W3: Motivation for studying chat models**
>
> > - Generality of the method.
> > - Quick Alignment Tax Study (Appendix F).
> > - Retraining with RL can be unstable due to noise in the reward model, sampling variability, and other factors. Training-time realignment acts as a tuning scale, guiding model trainers on where to realign for appropriate performance.
> >   - For instance, during the GPT-4o sycophancy incident (April 25, 2025), the team rolled back and retrained the model due to an inappropriate reward signal. **InRa can help identify the alignment tax, while TrRa may help prevent the need for retraining.**
> > - Subjective Alignment Preferences: People's views on 3H alignment vary—what one sees as sycophancy, another may find acceptable. **InRa allows adaptation to these differences. TrRa helps iterative products quickly.**
>
>
>
> **Q1: `When saying Llama 3.1 8B under the dialogue model, are you referring to chat models?`**
>
> > Yes, we will change these terms to chat models.
>
> **Q2: `I wonder if you did any ablation study on the impact of adapting different layer depths?`**
>
> > - In Section 5, we first analyze the relative importance of top versus bottom layers and find that **bottom layers are more influential**. We then apply adaptation to these layers and observe consistent performance, aligning with the earlier analysis. Adapting top layers often compromises language ability.
> >
> > - We also add the experiments on the **LLaMA3.2-3B** model, which has different layer depths compared to the 8B model, and observe **consistent results**.
> >
> > | LLama3.2-3B-Base | AlpacaEval2 | Arena-Hard | MT-Bench |
> > | ---------------- | ----------- | ---------- | -------- |
> > | top-1            | 4.03        | 4.00       | 5.85     |
> > | top-3            | 2.72        | 4.70       | 5.68     |
> > | bottom-1         | 9.86        | 9.80       | 6.53     |
> > | bottom-3         | 9.91        | 11.70      | 6.56     |

---

> > ### Comment · Reviewer_PCc3 · 2025-08-04
> >
> > Thank you for providing the additional experiment results and more detailed analysis. Your responses have addressed my concerns, and I will take this into consideration when revising my score.

---

> > > ### Author Response · Authors · 2025-08-05
> > > **Thank You for Your Feedback**
> > >
> > > We're glad our clarifications addressed your concerns.
> > >
> > > Thank you for your feedback~

---

### Official Review · Reviewer_WpBq · 2025-07-01

**Clarity:** 2
**Significance:** 2
**Originality:** 3
**Rating:** 3
**Confidence:** 4

**Summary:**

Within the problem space of aligning models, the authors use the DeRa (Decoding Time Realignment) technique to define a logit target that is interpolated or extrapolated from two models - in this case a DeepSeek-R1-Distilled-1.5B model and DeepScaleR-1.5B; they use this logit target to distill a model using a parameter-efficient finetuning technique (they use an extra layer added to the bottom, which they show to work better than simply using LoRA adapters), and this first step is called Training-time Realignment (TrRa). In a second contribution, the authors show that it is possible to apply TrRa iteratively to reach more extreme extrapolation with slightly less degradation.

In summary, the authors combine a number of ad-hoc techniques to get a similar effect as DeRa but with greater efficiency.

**Questions:**

Can you specify, in simple terms, what problem TrRa-iter solves in a more general case? Does it address use cases not already adressed by DeRa or is the main claim that you get better quality in extrapolation (noting that you're adding multiple layers at the same time if I understood the paper correctly)

**Ethical Concerns:**

["NO or VERY MINOR ethics concerns only"]

**Final Justification:**

While I am grateful to the authors for the clarifications and the effort they made to push the reviewers towards higher scores, I still think that the paper's contribution is too narrow and limited for a full NeurIPS paper.

**Limitations:**

In my opinion, the main limitation of the paper is the very one-dimensional framing: they take a problem that is relatively narrow and then show one recipe - made from known ingredients - for solving it. The limitations that the authors list (yes their way allows to load the original model only once, but you get two times the KV cache) are all technical limitations within this narrow framing which I find unhelpful

**Quality:**

3

**Strengths And Weaknesses:**

Strengths:
- The approach is described in sufficient detail for others to understand and possibly reimplement the approach
- Compared to the brute-force way of solving the same problem, the authors' approach is much less compute-heavy

Weaknesses:
- The paper does a very bad job of separating between the objective that the authors want to achieve and the way of achieving it. There is some evidence in the form of the LoRA ablation as well as the ablation of which layers to train and how to initialize them which suggests that the chosen appoach is indeed a good choice of a design to solve the problem the authors want to solve
- The problem as stated by the authors is too narrow to be practically relevant - in my opinion the goal of the paper should be more general to be relevant, even if that means that the paper doesn't reach that goal with the present work.
- The paper is somewhat incremental work on top of DeRa and DeepScaleR - Trading off solution quality versus length is a good problem to attack in general, but if that is the goal then the approach of the author only offers minimal novelty against what already exists

---

> ### Author Rebuttal · Authors · 2025-07-30
>
> **For your Summary**
>
> > We believe there may be some misunderstandings:
> >
> > (1). You mentioned, `“they use this logit target to distill a model using a parameter-efficient finetuning technique.”`
> >
> > In fact, TrRa can be implemented through both full fine-tuning and parameter-efficient fine-tuning techniques.
> >
> > (2). Additionally, it appears that the InRa method was overlooked. The layer adapter parameter-efficient finetuning technique is designed to facilitate the InRa method, which performs inference-time realignment.
> >
> > Please refer to Section 3.3 of our paper, where we discuss the relationship between TrRa and InRa.
> >
> > (3). Upon careful consideration of your review, we realized you may have missed that we also extend our method to the 3H experiments.
> >
> > ---
> >
> > **To aid your understanding, we provide the following summary:**
> >
> > We propose training-time realignment (TrRa) and inference-time realignment (InRa) to better combine the logits of a base model and an aligned model for controllable alignment. Traditional methods for determining the optimal regularization level require retraining multiple models with varying strengths. TrRa addresses this limitation by efficiently identifying where to adjust the model using limited computational resources—without the uncertainty and cost of retraining-based methods. InRa offers an API interface, similar to temperature, allowing users to customize regularization strength during generation.
>
> **W1: Misunderstading**
>
> > This point may originate from your misunderstanding summary; you may neglect the layer adapter is to faciliate the InRa. Our TrRa is full fine tuning.
> >
> > **But thank you for acknowledging our chosen appoach is indeed a good choice.**
>
> **W2: `The paper should be more general to be relevant, even if that means that the paper doesn't reach that goal with the present work.`**
>
> > (1) In Section 4.3, we further extend our method to chat models, enabling the **adjustment of 3H values** and demonstrating the generality of our approach.
> >
> > (2) In Section 5.2 (Hyperparameter Stability), we show that the hyperparameter setting in our method aligns closely with that of full fine-tuning, suggesting that our approach can serve as **a lightweight proxy for hyperparameter tuning**. **This is the practically relevant claim of DeRa.**
> >
> > (3) In Appendix F.1, we demonstrate that our method can also facilitate the study of **alignment tax**.
> >
> > (4) Moreover, with the rapid development of reasoning models, both **efficient thinking** (e.g., **DeepScaleR**) and **adaptive thinking** (e.g., hybrid reasoning in **Claude 4**, and the thinking budget mechanism in **Gemini 2.5**) are gaining importance. We argue that the two scenarios we consider—efficient and adaptive thinking—are highly relevant in practice, **as they contribute to user satisfaction and reduce inference costs.**
> >
> > (5) TrRa reduces the need to retrain models from scratch, introducing a new paradigm for post-training alignment.
>
> **W3 : `Trading off solution quality versus length is a good problem to attack in general, but if that is the goal then the approach of the author only offers minimal novelty against what already exists`**
>
> > **Thank you for acknowledging that trading off solution quality versus length is an important problem. We would like to highlight that our work offers several novel contributions beyond this specific aspect:**
> >
> > - **Generality across modalities:**
> >   In Section 4.3, we extend our method to chat models, allowing adjustment of 3H alignment values , thereby demonstrating the broad applicability of our approach.
> > - **Efficiency and performance:**
> >   While DeepScaleR demonstrates strong performance and flexible control, it requires **3,800 A100 GPU hours** according to their blog. In contrast, our TrRa approach achieves **better performance** at **only 4 A100 hours** of training (see Table 1, particularly at λ = 1.5), highlighting a significant improvement in cost-effectiveness.
> > - **Beyond TrRa and InRa:**
> >   Our paper also provides additional insights and technical contributions:
> >   - We conduct an in-depth analysis and find that **bottom layers are particularly influential** for alignment (see Section 5.1).
> >   - We propose a **novel parameter-efficient training technique** using a single-layer adapter, enabling smooth inference-time realignment without full fine-tuning.
> > - **Comparison to DeRa:**
> >   see Q 1.
>
> **Q1:**
>
> >  `What problem TrRa-iter solves in a more general case?`
> >
> > - **TrRa-iter** further validates our algorithm and demonstrates that the reward signal can be **effectively transferred**. TrRa-iter solves the general problem of scalable, training-time realignment toward arbitrary alignment targets — especially when those targets lie beyond what DeRa or a single TrRa step can reach.
> >
> > - It does this by iteratively updating the model using intermediate, logit-fused teacher distributions, without the cost of full retraining.
> >
> >  `Does it address use cases not already adressed by DeRa?`
> >
> >   DeRa is indeed a strong method, but there remain areas for improvement. For example, as reasoning models evolve and generate increasingly **long chains of thought, the cost of DeRa becomes less acceptable due to its runtime overhead.** Our TrRa addresses this by enabling efficient training-time realignment. In addition, InRa allows alignment information to be injected with minimal training resources—using just a single-layer adapter—which is not feasible with DeRa.
> >
> >   | Difference             | DeRa                                                 | TrRa                                                         | InRa                                                         |
> >   | ---------------------- | ---------------------------------------------------- | ------------------------------------------------------------ | ------------------------------------------------------------ |
> >   | Stage                  | Inference-time                                       | Training-time                                                | Inference-time                                               |
> >   | Updates original model weights? | No                                                   | Yes                                                          | No                                                           |
> >   | Mechanism              | Fuses reference/aligned logits at each decoding step | Learns from interpolated logits as teacher (like knowledge distillation) | Inserts trainable adapter layer before LM layers and fuses outputs |
> >   | Iterative realignment? | No                                                   | Yes                                                          | No                                                           |
> >   | Inference cost         | double forward passes and doubled KV-cache           | No changes                                                   | Slightly higher (adapter + original layer + extra KV-cache)  |
> >   | Model depolyment       | Two Models                                           | No changes                                                   | Single model with integrated adapter layer                   |
> >   | User control?          | Yes – λ is tunable during inference                  | not tunable after deployment                                 | Yes – λ is tunable during inference                          |
> >   | Trainable component    | None                                                 | Whole model                                                  | Only bottom-layer adapter                                    |
> >
> >  ` (noting that you're adding multiple layers at the same time if I understood the paper correctly)`
> >
> >   Just to clarify, the adapter we use is a **single** additional layer (not multiple layers), inserted before the original bottom layer.
>
> **We sincerely hope this addresses your concerns and that you might consider revising your score accordingly.**

---

> ### Comment · Reviewer_WpBq · 2025-08-05
>
> Thank you for the extensive response. Let me react to each of the points
>
> First point - yes it can also be used for full fine-tuning. However in the experiments for this paper your recipe includes parameter-efficient tuning (and the InRa technique later in the paper depends on it). [cf W1 and the potential misunderstanding]
>
> ad (3) - indeed I missed the second experiment of studying the alignment tax. Thank you for the clarification.
>
> re W1 - please help me understand this as it doesn't seem to be explained well in the paper - in your comment you claim that TrRa is full finetuning, which would then mean that adding the adapter layer and finetuning it is not part of TrRa but happens elsewhere? Or is the claim that TrRa COULD be done as full fine-tuning if one didn't want to combine it with InRa [assuming we are to take the "Inference-time" to mean that no additional training happens in InRa]?
>
> re W2 - my statement was meant to convey: the goal in this paper is to mitigate overalignment, which is relatively narrow. Agree that the authors did a good job of finding actual examples in the wild, and that it's perceivable that a practical solution has relevance for some users.
>
> re W3 - Agree that the main contribution of this paper is greater efficiency, to the point of making the approach practical in some contexts where the baseline approach would be prohibitive (hence I don't understand the criticism of optimizing the \lambda in the other reviews). Put more succinctly the criticism here is that the TrRa/InRa combination is an ad-hoc solution that narrowly solves the over-/underalignment situation (which the authors call "general across modalities").
>
> ad Q1 - thank you for making explicit the differences between DeRa and the TrRa/InRa combination and that the main concern is efficiency.

---

> > ### Author Response · Authors · 2025-08-06
> > **Looking Forward to Your Feedback**
> >
> > Thank you for your time again!
> >
> > (1) `First point`:
> >
> > >  Concerns Solved. See (3).
> >
> > (2)  `ad (3)`:
> >
> > > Concerns Solved.
> >
> > (3) `Re W1`
> >
> > > In our paper, the use of the adapter layer occurs within InRa (see Section 3.2). Your second point is correct.  We will include a more detailed description of both usages in the Method section.
> > >
> > > See (5)
> >
> > (4) `Re W2`:
> >
> > > Our method is designed not only to mitigate overalignment, **but also to address underalignment.** The experiments presented in our paper support this claim. See the following axes for a visualization of the effect of $\lambda$.
> > >
> > > ```
> > >  |--------------|--------------|------------|-------------------|------>
> > >  0.0            0.5            1.0           2~5               >5.0
> > >  (Base)         ↓    (already aligned)  (More aligned)   ⚠️ Reward Hacking Risk
> > >                 |
> > >        Base → Less Aligned
> > > ```
> > >
> > > Thanks for your acknowledgment!
> >
> > (5) `Re W3`:
> >
> > > - I guess that other reviewers may focus more on the performance rather than the flexibility provided by $\lambda$, or the efficiency and scalability of alignment strength adjustment. Because we have also achieved strong performance gains on reasoning tasks.
> > > - **Both methods can be used independently**, as their designs are grounded in the theory presented in Equations (1)–(5). **Combining them is not necessary**.
> > > - **TrRa is an objective** that can be integrated with various training strategies, such as full fine-tuning, LoRA, or our proposed layer adapter.
> > > - In contrast, **InRa requires the layer adapter to inject alignment information** through training objectives such as SFT, DPO, or TrRa, as noted in `Line 157` of our paper. **Besides, the layer adapter facilitates adaptive control of alignment strength during inference.**
> > > - Section 4.1 presents TrRa with full fine-tuning. Section 4.2 applies SFT on the layer adapter, and Section 4.3 applies DPO on the layer adapter.
> >
> > (6)` re Q1`:
> >
> > > Concerns Solved. TrRa has no efficiency concern, and InRa achieves great efficiency compared to DeRa.
> >
> > **If you have any further concerns, please feel free to reach out. We sincerely hope our responses address your concerns, and we would greatly appreciate it if you would consider revising your score.**

---

> > ### Author Response · Authors · 2025-08-08
> > **Looking Forward to Your Feedback**
> >
> > Hi Reviewer WpBq,
> >
> > Sorry to trouble you again. We sincerely hope that the discussion at NeurIPS can proceed constructively, as silence cannot resolve misunderstandings.
> >
> > If you have any concerns, we would be glad to discuss them and provide clarification within the remaining time.
> >
> > Best regards,
> >
> > The Authors

---

### Official Review · Reviewer_L9eG · 2025-07-03

**Clarity:** 2
**Significance:** 2
**Originality:** 3
**Rating:** 4
**Confidence:** 3

**Summary:**

This paper proposes training-time realignment (TrRa) and inference-time realignment (InRa) to better combine the logits of base model and reference models for control the alignment. TrRa can achieve similar or better MATH500/AIME24 performance while shorten the response length half.

**Questions:**

The DeepSeek-R1-Distill-Qwen-1.5B's MATH500 score is 80.23 in the paper, while 83.9 in its official report. Is that difference caused by temperature difference? If so, when utilizing the same settings as official report, what would TrRa and InRa perform?

**Ethical Concerns:**

["NO or VERY MINOR ethics concerns only"]

**Final Justification:**

I would keep my current score as reviewer solve most of my concerns

**Limitations:**

authors mention it in Appendix A

**Quality:**

3

**Strengths And Weaknesses:**

Strengths:

1. The idea is clear and the writing is easy to folllow.
2. The performance of TrRa is good, achieve similar or better MATH500/AIME24 performance while shorten the response length half. And the computation cost of TrRa is not high (200 step finetuning). InRa is also more flexible.
3. Trying different benchmarks and models to show the effectiveness

Weakness:
1. It seems requiring to search the best hyperparameter lambda for getting the best performance. And there are also some other parameters can be tuned (like temperature), it's not clear that if the best lambda is different for different temperature
2. may decrease the throughput and increase the cache in inference. And if training TrRa for more iterations, the performance may drop.
3. could move Fig.2 to appendix because it's common for the readers of this paper. Maybe Fig3 can using caption with TrRa and InRa to be more clear.

---

> ### Author Rebuttal · Authors · 2025-07-30
>
> #
>
> ```
>  |-----------|-----------|-------------------|-----------|------>
>  0.0         0.5        1.0                 2~5          >5.0
>  (Base)       ↓      (already aligned)   (More aligned)  ⚠️ Reward Hacking Risk
>               |
>         Base → Aligned
> ```
>
> **W1: Function of $\lambda$**
>
> > - Table 1 in our paper illustrates how different $\lambda$ values in TrRa enable **flexible** training-time realignment. The #Token column aligns well with the above axes, **showing where to adjust the model** without the instability of retraining methods like RL, which often suffer from sampling and other factors.
> > - In InRa, $\lambda$ functions similarly to a temperature parameter: larger values lead to stronger alignment, while smaller values result in weaker alignment. It serves as a **customizable** option to enable adaptive reasoning **based on the user's budget.** It is a parameter that can be customized during the inference stage.
> > - For the effect of temperature, see **Q1**.
> >
> > Overall, adjustments on the Pareto plane prioritize trade-offs rather than optimizing for a single best outcome.
>
> **W2: Limiation**
>
> > - Indeed, InRa faces this issue, as acknowledged in the limitations. We hope that KV compression or other engineering optimizations can help mitigate this problem. But TrRa does not suffer from this problem.
> >
> > - Performance drop with more iterations is inevitable.  TrRa-iter demonstrates that the reward signal can be effectively **transferred** and the correctness of TrRa.
> >   - Suppose you use λ=2 in iteration 1, resulting in a better-aligned model.
> >   - If you then use λ=2 again in iteration 2, it may be equivalent to λ>4 in iteration 1 (based on the empirical experiments).
> >   - Therefore, due to **reward hacking**, performance degradation becomes **unavoidable** with more iterations of TrRa.
>
> **W3: Thanks for your suggestion!**
>
> > We fully accept your suggestion and make the corresponding revisions.
>
>
>
> **Q1: The temperature effect on the performance is negligible.**
>
> > - The evaluation difference between ours and DeepSeek is as follows:
> >
> > |             | Temperature | Generation Length | Average times |
> > | ----------- | ----------- | ----------------- | ------------- |
> > | DeepSeek-R1 | 0.6         | 32796             | 64            |
> > | TrRa        | 0.7         | 16483             | 8             |
> >
> > - We think the results difference **depends on reasoning tokens**; therefore, we extend the generation length to 32K, and the results are as follows:
>
> **For TrRa:**
>
> > | DeepSeek-R1-Distilled-1.5B | Math500 (Avg@8) | Token |
> > | -------------------------- | --------------- | ----- |
> > | temperature = 0.6          | 84.95           | 5437  |
> > | temperature = 0.7          | 85.38           | 5219  |
> >
> > - It is **consistent** with the DeepSeek report.
> >
> > | TrRa $\lambda=0.5$ | Math500 (Avg@8) | Token |
> > | ------------------ | --------------- | ----- |
> > | temperature = 0.6  | 88.25           | 3936  |
> > | temperature = 0.7  | 87.88           | 3949  |
> >
> > | TrRa $\lambda= 2$ | Math500 (Avg@8) | Token |
> > | ----------------- | --------------- | ----- |
> > | temperature = 0.6 | 89.10           | 2859  |
> > | temperature = 0.7 | 89.35           | 2862  |
> >
> > - Our method achieves more token reduction and high performance.
>
> **For InRa:**
>
> > Our customized implementation VLLM engine results are as follows:
> >
> > | InRa $\lambda= 0.0$ | Math500 (Avg@8) | Token |
> > | ------------------- | --------------- | ----- |
> > | temperature = 0.6   | 84.53           | 5543  |
> > | temperature = 0.7   | 85.88           | 5289  |
> >
> > | InRa $\lambda= 0.5$ | Math500 (Avg@8) | Token |
> > | ------------------- | --------------- | ----- |
> > | temperature = 0.6   | 72.60           | 2524  |
> > | temperature = 0.7   | 71.43           | 2224  |
> >
> > | InRa $\lambda= 1.0$ | Math500 (Avg@8) | Token |
> > | ------------------- | --------------- | ----- |
> > | temperature = 0.6   | 64.75           | 461   |
> > | temperature = 0.7   | 64.38           | 459   |
> >
> > - When $\lambda = 0$, the result is consistent with the DeepSeek report. By increasing $\lambda$, users can flexibly trade off between performance and token reduction based on their budget.

---

> ### Comment · Reviewer_L9eG · 2025-08-07
>
> thanks for the response, I would keep my current positive score

---

> > ### Author Response · Authors · 2025-08-07
> > **Thank You for Your Positive Feedback**
> >
> > Thank you for your positive feedback.

---

> > > ### Author Response · Authors · 2025-08-08
> > > **Supplementary Note**
> > >
> > > **Regarding W2, we provide the following supplementary note: InRa achieves significantly higher efficiency compared to the DeRa method with comparable performance.**
> > >
> > >
> > > | Difference             | DeRa                                                 | TrRa                                                         | InRa                                                         |
> > > | ---------------------- | ---------------------------------------------------- | ------------------------------------------------------------ | ------------------------------------------------------------ |
> > > | Stage                  | Inference-time                                       | Training-time                                                | Inference-time                                               |
> > > | Updates model weights? | No                                                   | Yes                                                          | No                                                           |
> > > | Mechanism              | Fuses reference/aligned logits at each decoding step | Learns from interpolated logits as teacher (like knowledge distillation) | Inserts trainable adapter layer before LM layers and fuses outputs |
> > > | Iterative realignment? | No                                                   | Yes                                                          | No                                                           |
> > > | Inference cost         | double forward passes and doubled KV-cache           | No changes                                                   | Slightly higher (adapter + original layer + extra KV-cache)  |
> > > | Model depolyment       | Two Models                                           | No changes                                                   | Single model with integrated adapter layer                   |
> > > | User control?          | Yes – λ is tunable during inference                  | not tunable after deployment                                 | Yes – λ is tunable during inference                          |
> > > | Trainable component    | None                                                 | Whole model                                                  | Only bottom-layer adapter                                    |

---

### Note · Authors · 2025-08-14

- We addressed all concerns raised by reviewers PCc3 and L9eG, gaining their positive support and further improving our paper.

- We believe we have also resolved reviewer WPBq’s misunderstandings, contradictions, and oversight of the main experiments.

- We appreciate the valuable discussion.

- We also provided the discussion summary in the confidential comments earlier.

---

### Decision · Program_Chairs · 2025-09-17

**Decision:**

Accept (poster)

**Comment:**

This paper proposes a flexible realignment framework for language models, introducing both training-time realignment (TrRa) and inference-time realignment (InRa). The approach enables controllable alignment by fusing logits at different stages, significantly reducing token usage while maintaining or even improving reasoning performance across benchmarks such as MATH500 and AIME24. Reviewers generally praised the clarity of writing, technical soundness, and the breadth of experiments. Reviewers raised concerns about performance degradation in some settings and further discussion of motivation in applications. The rebuttal addressed some main concerns. Although one reviewer still remains skeptical, I regard their concerns to be mainly about scope but not technical matters. Therefore, the merits still outweigh weaknesses. I recommend to accept this paper.